# Generative multitask learning mitigates target-causing confounding

**Taro Makino**[1]   **Krzysztof J. Geras**[2,1]   **Kyunghyun Cho**[1,3,4]

[1]NYU Center for Data Science
[2]NYU Grossman School of Medicine
[3]Genentech   [4]CIFAR LMB
taro@nyu.edu

## Abstract

We propose generative multitask learning (GMTL), a simple and scalable approach to causal machine learning in the multitask setting. Our approach makes a minor change to the conventional multitask inference objective, and improves robustness to target shift. Since GMTL only modifies the inference objective, it can be used with existing multitask learning methods without requiring additional training. The improvement in robustness comes from mitigating unobserved confounders that cause the targets, but not the input. We refer to them as *target-causing confounders*. These confounders induce spurious dependencies between the input and targets. This poses a problem for conventional multitask learning, due to its assumption that the targets are conditionally independent given the input. GMTL mitigates target-causing confounding at inference time, by removing the influence of the joint target distribution, and predicting all targets jointly. This removes the spurious dependencies between the input and targets, where the degree of removal is adjustable via a single hyperparameter. This flexibility is useful for managing the trade-off between in- and out-of-distribution generalization. Our results on the Attributes of People and Taskonomy datasets reflect an improved robustness to target shift across four multitask learning methods.

## 1   Introduction

Deep neural networks (DNNs) excel at extracting patterns from unstructured data, but these patterns can fail to generalize outside of the training distribution. These failures are often attributed to DNNs learning statistical associations rather than causal relations [Ribeiro et al., 2016, Jo and Bengio, 2017, Geirhos et al., 2020]. Therefore, there is a concerted effort to make patterns learned by DNNs satisfy certain properties of causal relations, such as invariance and modularity [Schölkopf, 2019, Schölkopf et al., 2021, Wang and Jordan, 2021]. This research direction is called causal machine learning (ML). Existing approaches to causal ML often require additional information in the form of labeled environments [Arjovsky et al., 2019, Lu et al., 2021] or labeled confounders [Puli et al., 2021]. These requirements are restrictive, and it is therefore beneficial to identify more natural settings where existing information can be used for causal ML.

We consider the setting of multitask learning (MTL) [Caruana, 1997, Ruder, 2017, Zhang and Yang, 2017], where there is an input $\mathbf{x} \in \mathcal{X}$ and multiple targets. Without loss of generality, we consider two targets $\mathbf{y} \in \mathcal{Y}$ and $\mathbf{y}' \in \mathcal{Y}'$. As is common in the MTL literature, we distinguish $\mathbf{y}$ as being the main target, and $\mathbf{y}'$ as the auxiliary target. Both targets are used during training, but only the main target is used during evaluation. In the conventional approach to MTL, which we call *discriminative*

36th Conference on Neural Information Processing Systems (NeurIPS 2022).

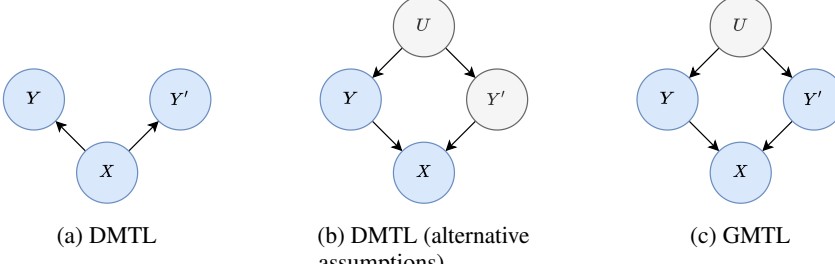

(a) DMTL

(b) DMTL (alternative assumptions)

(c) GMTL

Figure 1: (a) The conventional approach to MTL, which we call discriminative multitask learning (DMTL), assumes the targets $\mathbf{y}$ and $\mathbf{y}'$ are conditionally independent given the input $\mathbf{x}$. Due to this assumption, the inference objective for predicting the main target $\mathbf{y}$ is $\operatorname{argmax}_{\mathbf{y}} \log p(\mathbf{y} \mid \mathbf{x})$. (b) DMTL is flawed under an alternative set of assumptions, where the targets cause the input, and there exists an unobserved confounder $\mathbf{u}$ that causes the targets, but not the input. We call $\mathbf{u}$ a target-causing confounder. Since $\mathbf{y}'$ is unobserved when predicting $\mathbf{y}$ with DMTL, this unblocks the backdoor path $\mathbf{y} \leftarrow \mathbf{u} \rightarrow \mathbf{y}' \rightarrow \mathbf{x}$ and makes the input and targets spuriously dependent. That is, $p(\mathbf{y} \mid \mathbf{x})$ shifts when $p(\mathbf{u})$ shifts. (c) We propose generative multitask learning (GMTL), where the inference objective is $\operatorname{argmax}_{\mathbf{y}, \mathbf{y}'} \log p(\mathbf{x} \mid \mathbf{y}, \mathbf{y}')$. Unlike DMTL, this objective conditions on all targets, which $d$-separates $\mathbf{x}$ and $\mathbf{u}$, and removes the spurious dependencies between the input and targets.

*multitask learning* (DMTL), the inference objective is

$$\operatorname*{argmax}_{\mathbf{y}, \mathbf{y}'} \log p(\mathbf{y}, \mathbf{y}' \mid \mathbf{x}) = \operatorname*{argmax}_{\mathbf{y}, \mathbf{y}'} \log p(\mathbf{y} \mid \mathbf{x}) + \log p(\mathbf{y}' \mid \mathbf{x}). \tag{1}$$

This factorization corresponds to assuming the targets are conditionally independent given the input. This is a convenient assumption to make, since it prevents $\mathcal{Y} \times \mathcal{Y}'$ from growing exponentially with the number of tasks. Since only the main target $\mathbf{y}$ is used during evaluation, the inference objective is

$$\operatorname*{argmax}_{\mathbf{y}} \log p(\mathbf{y} \mid \mathbf{x}). \tag{2}$$

DMTL does not make causal assumptions explicitly, but its conditional independence assumption is consistent with the causal graph in Fig. 1a. We argue that DMTL is flawed under an alternative set of assumptions.

First, we assume the targets cause the input, which implies we are predicting the cause from the effect. This is called anticausal learning, and is considered to describe many common problems such as image classification [Schölkopf et al., 2012]. For example, in object recognition, the object category causes the pixel representation of that object. Second, we assume there exists a variable $\mathbf{u}$ that causes the targets, but not the input. We call $\mathbf{u}$ a *target-causing confounder*.

The problem with DMTL under these alternative assumptions is that the auxiliary target $\mathbf{y}'$ is unobserved, as illustrated in Fig. 1b. This is problematic because it unblocks the backdoor path $\mathbf{y} \leftarrow \mathbf{u} \rightarrow \mathbf{y}' \rightarrow \mathbf{x}$, and makes $\mathbf{x}$ and $\mathbf{y}$ spuriously dependent. Since the backdoor path passes through $\mathbf{u}$, DMTL is sensitive to shifts in $p(\mathbf{u})$. When $p(\mathbf{u})$ shifts, the relationship between $\mathbf{x}$ and $\mathbf{y}$ changes, and DMTL fails to generalize. This is the weakness of DMTL that we aim to improve. Since $\mathbf{u}$ causes the targets, a shift in $p(\mathbf{u})$ leads to a shift in $p(\mathbf{y}, \mathbf{y}')$, which is often called target shift [Zhang et al., 2013]. The targets are observable, unlike $\mathbf{u}$, so we use target shift as an observable proxy for shifts in $p(\mathbf{u})$.

We propose *generative multitask learning* (GMTL) as a way to mitigate target-causing confounding. GMTL is based on the following idea. If we write the causal factorization of the joint distribution of the observed variables as

$$p(\mathbf{x}, \mathbf{y}, \mathbf{y}') = p(\mathbf{x} \mid \mathbf{y}, \mathbf{y}')p(\mathbf{y}, \mathbf{y}') = p(\mathbf{x} \mid \mathbf{y}, \mathbf{y}') \int p(\mathbf{y} \mid \mathbf{u})p(\mathbf{y}' \mid \mathbf{u})p(\mathbf{u})\mathrm{d}\mathbf{u},$$

this shows that $p(\mathbf{x} \mid \mathbf{y}, \mathbf{y}')$ is invariant to shifts in $p(\mathbf{u})$. Therefore, if we use

$$\operatorname*{argmax}_{\mathbf{y}, \mathbf{y}'} \log p(\mathbf{x} \mid \mathbf{y}, \mathbf{y}') \tag{3}$$

as the inference objective, we become invariant to shifts in $p(\mathbf{u})$. Importantly, unlike Eq. 1, the inference objective in Eq. 3 no longer factorizes over the targets. This means that even if we only care about evaluating the main target $\mathbf{y}$, we also need to consider $\mathbf{y}'$. Conditioning on all targets during inference closes the backdoor paths between the inputs and targets. This $d$-separates $\mathbf{x}$ and $\mathbf{u}$, and removes the spurious dependencies between the input and targets.

In practice, we implement GMTL as an approximation of Eq. 3. As we discuss in Section 3, this allows us to formulate GMTL as a minor change to the inference objective of DMTL. This makes GMTL practical, since it can be used with existing MTL methods without requiring additional training. We also introduce a single hyperparameter to adjust the degree to which we remove spurious dependencies between the input and targets. This makes GMTL include DMTL as a special case, where no spurious dependencies are removed. This flexibility is useful for managing the trade-off between in-distribution (ID) and out-of-distribution (OOD) generalization, since spurious dependencies are predictive in the former setting, but not the latter.

In order to empirically validate GMTL, we perform experiments on two datasets called Attributes of People [Bourdev et al., 2011] and Taskonomy [Zamir et al., 2018]. Our results show that GMTL improves robustness to target shift across four MTL methods. We attribute this improvement to GMTL's ability to mitigate target-causing confounding. This serves as evidence that causal ML shows promise for MTL.

## 2 Background

### 2.1 Out-of-distribution generalization

OOD generalization refers to the capability of ML systems to generalize on data outside of the training distribution. Bengio et al. [2021] describe it as one of the greatest challenges in ML, and one that cannot be solved solely by scaling up datasets and computation. The failure of ML systems to generalize OOD is a general problem that has been reported across a wide range of problems [D'Amour et al., 2020]. The core issue is that it is often easier for ML systems to learn patterns that are dataset-specific, rather than those that hold universally, and there is often no incentive to learn the latter. These dataset-specific patterns have many names, such as surface statistical regularities [Jo and Bengio, 2017], nonrobust features [Tsipras et al., 2019], and shortcuts [Geirhos et al., 2020]. We take a causal perspective and adopt the term *spurious dependencies*, which are defined as statistical dependencies not supported by causal links [Pearl, 2009].

### 2.2 Dataset biases induce spurious dependencies

Spurious dependencies can arise from design choices in dataset creation. Recht et al. [2019] demonstrated that state-of-the-art image classifiers trained on ImageNet [Deng et al., 2009] failed to generalize to a replicated test set designed to mirror the original test set distribution. Engstrom et al. [2020] offered an explanation for this phenomenon. The authors reported that the dataset replication procedure relied on estimating a human-in-the-loop metric called selection frequency, and that bias in estimating this statistic led to undesirable variation across datasets. Selection frequency is defined for an input-target pair, and measures the proportion of crowdsourced annotators who deem the pair correctly labeled. Recht et al. [2019] performed statistic matching on this metric, which corresponds to conditioning on it. Since selection frequency is caused by the input and target, this corresponds to conditioning on a collider, which induces a spurious dependency between the input and target. Such a dependency would fail to generalize OOD, which may explain why predictive performance degraded on the replicated test set.

In this work, we address a different source of dataset bias that we call target-causing confounding. A variable is a target-causing confounder if it is causes the targets, but not the input. For a simple example, suppose we have images of people in indoor scenes, and the targets are clothing-related attributes such as a hat and scarf. The season causes the targets, since people pair various types of clothing depending on the season. The season does not cause the image, since the images are of indoor scenes. Therefore, the season is a target-causing confounder that induces spurious dependencies between the input and targets. Relying on such dependencies for prediction can be problematic if the training and test sets represent different seasons. Although this is a simple and contrived example

for the purpose of illustration, dataset biases can occur in real-life settings like medicine, and have serious societal impacts [Mehrabi et al., 2021].

## 2.3 Causal machine learning

In Section 2.1, we made the distinction between patterns that are dataset-specific, and those that hold universally. The field of causality formalizes the notion of patterns that hold universally, calling them causal relations [Pearl, 2009]. Causal ML aims to use ideas from causality to improve machine learning methods, particularly in the area of OOD generalization [Zhang et al., 2013, Arjovsky et al., 2019, Schölkopf, 2019, Wang and Blei, 2019, Puli et al., 2021, Schölkopf et al., 2021]. Our work falls under this category, since GMTL improves robustness to target shift by using an intervention distribution for prediction. This idea is also shared by Subbaswamy et al. [2019], who propose a general framework for estimating intervention distributions for robust prediction. GMTL is a practical method for achieving this on high-dimensional datasets. Relatedly, Makar et al. [2022] use importance reweighting to learn distributions that are invariant to intervention on an auxiliary target. Both Subbaswamy et al. [2019] and Makar et al. [2022] assume the same causal graph as GMTL.

## 2.4 Multitask learning

MTL can reduce computational cost through parameter sharing, and improve predictive performance over training on each task individually [Standley et al., 2020]. Its use is widespread, spanning domains such as scene understanding [Zamir et al., 2018, Chowdhuri et al., 2019, Zamir et al., 2020], medical diagnosis [Kyono et al., 2021], natural language processing [McCann et al., 2018, Liu et al., 2019, Radford et al., 2019, Wang et al., 2019, Worsham and Kalita, 2020, Aghajanyan et al., 2021], recommender systems [Covington et al., 2016, Ma et al., 2018], and reinforcement learning [Kalashnikov et al., 2021]. Distinguishing between main and auxiliary targets is common [Ruder, 2017, Liebel and Körner, 2018, Vafaeikia et al., 2020], but not universal, and is primarily important for concerns such as which loss to use for early stopping [Caruana, 1997]. We make this distinction throughout this paper, denoting the main target as $\mathbf{y}$, and the auxiliary target as $\mathbf{y}'$.

# 3 Generative multitask learning

In GMTL, we condition on all targets in order to $d$-separate $\mathbf{x}$ and $\mathbf{u}$, and remove spurious dependencies between the input and targets. During inference, we rely solely on $p(\mathbf{x} \mid \mathbf{y}, \mathbf{y}')$, since is invariant to shifts in $p(\mathbf{u})$. Approaching this naively by estimating $p(\mathbf{x} \mid \mathbf{y}, \mathbf{y}')$ directly can be difficult, especially when $\mathbf{x}$ is high-dimensional. We can get around this by using Bayes' rule to write

$$\operatorname*{argmax}_{\mathbf{y},\mathbf{y}'} \log p(\mathbf{x} \mid \mathbf{y}, \mathbf{y}') = \operatorname*{argmax}_{\mathbf{y},\mathbf{y}'} \log \frac{p(\mathbf{y}, \mathbf{y}' \mid \mathbf{x})p(\mathbf{x})}{p(\mathbf{y}, \mathbf{y}')}$$
$$= \operatorname*{argmax}_{\mathbf{y},\mathbf{y}'} \log p(\mathbf{y}, \mathbf{y}' \mid \mathbf{x}) - \log p(\mathbf{y}, \mathbf{y}'). \tag{4}$$

This shows that the argmax over the targets results in the difficult-to-estimate $p(\mathbf{x})$ term to drop out, meaning we only need to estimate $p(\mathbf{y}, \mathbf{y}' \mid \mathbf{x})$ and $p(\mathbf{y}, \mathbf{y}')$.

In order to make GMTL practical, we replace Eq. 4 with an approximation, i.e.

$$\operatorname*{argmax}_{\mathbf{y},\mathbf{y}'} \log p(\mathbf{y} \mid \mathbf{x}) + \log p(\mathbf{y}' \mid \mathbf{x}) - \alpha \log p(\mathbf{y}, \mathbf{y}'), \qquad \alpha \in [0, 1]. \tag{5}$$

This introduces two changes. The first is that we allow $p(\mathbf{y}, \mathbf{y}' \mid \mathbf{x})$ to factorize, as was the case in DMTL. This allows us to use conventional approaches to estimate $p(\mathbf{y}, \mathbf{y}' \mid \mathbf{x})$, which is beneficial since training is notoriously difficult in MTL [Standley et al., 2020]. To be clear, we do not allow this factorization because we assume the targets are conditionally independent given the input, as this would violate our causal graph in Fig. 1c. We do so purely out of convenience, so that GMTL can leverage existing approaches for estimating $p(\mathbf{y}, \mathbf{y}' \mid \mathbf{x})$. We can allow this factorization as long as we capture the dependency between the targets when estimating $p(\mathbf{y}, \mathbf{y}')$. This prevents Eq. 5 from factorizing over the targets, and satisfies our requirement of conditioning on all targets in order to remove spurious dependencies between the input and targets. We discuss some complications with estimating $p(\mathbf{y}, \mathbf{y}')$ in Section 3.2, but in general, it can be done trivially for categorical targets.

Therefore, GMTL only changes the inference objective of DMTL, and can be applied with existing multitask methods without requiring additional training.

The second change is the introduction of the hyperparameter $\alpha \in [0, 1]$, which enables us to interpolate between DMTL ($\alpha = 0$) and the most extreme case of GMTL ($\alpha = 1$). $\alpha$ controls the degree to which we remove spurious dependencies between the input and targets. The ability to adjust this effect is important because if the training and test distributions are identical, then we should be willing to use any dependencies available, regardless of whether they are spurious or causal. However, it is more realistic to assume training and test distributions will be different, and under these circumstances, setting $\alpha > 0$ can improve robustness to target shift.

## 3.1 Interpreting $\alpha$

The hyperparameter $\alpha$ represents our uncertainty about the test target distribution. In the case of complete certainty, i.e. if we know that the test target distribution is identical to the empirical target distribution, we set $\alpha = 0$ to use the empirical distribution. In contrast, in the case of complete uncertainty, we set $\alpha = 1$ to completely remove the influence of the empirical distribution. This is equivalent to assuming the test target distribution is uniform. Generally, it can be shown that GMTL corresponds to replacing the empirical target distribution $p(\mathbf{y}, \mathbf{y}')$ with a distribution proportional to $p(\mathbf{y}, \mathbf{y}')^{1-\alpha}$. The derivation is in the supplementary material. We perform this replacement during inference in order to keep training the same as DMTL. In principle, we could achieve the same invariance by reweighting examples by $1/p(\mathbf{y}, \mathbf{y}')^{\alpha}$ during training [Zhang et al., 2013, Makar et al., 2022].

This raises an important point - $\alpha$ represents an assumption, and is not a hyperparameter that can be tuned in the conventional sense. Instead, tuning $\alpha$ is analogous to model selection w.r.t. an unknown distribution, which is a difficult open problem [Gulrajani and Lopez-Paz, 2021]. Progress in this direction is likely to benefit GMTL, as well as other methods with similar hyperparameters [Wortsman et al., 2022]. In the remainder of this work, we assume oracle access to the optimal $\alpha$ for a given OOD test distribution. That is, our main focus is not on the difficulty of tuning $\alpha$, which is a separate research question. Instead, we study the efficacy of GMTL assuming we have correctly assessed how similar the training and test target distributions are, i.e. knowing the optimal $\alpha$.

Having said that, we provide a simple and effective heuristic for setting $\alpha$ even for an unknown test distribution, which makes GMTL practical. This is based on the fact that $\alpha$ represents a trade-off. Increasing $\alpha$ improves target shift robustness at the expense of ID predictive performance. We show empirically in Section 6 that this relationship is very strong, and holds across two datasets, multiple pairs of tasks, and four MTL methods. Therefore, our heuristic is to set $\alpha$ to the maximum value within an acceptable loss in ID predictive performance, where the latter is measured using the validation set. This ensures that we are as robust as possible to target shift, while keeping ID predictive performance at an acceptable level. These results are in the supplementary material.

If we do have access to the test target distribution $q(\mathbf{y}, \mathbf{y}')$, then we can do away with $\alpha$ and simply replace $p(\mathbf{y}, \mathbf{y}')$ with $q(\mathbf{y}, \mathbf{y}')$, which results in the following inference objective:

$$\underset{\mathbf{y},\mathbf{y}'}{\arg\max} \log p(\mathbf{y} \mid \mathbf{x}) + \log p(\mathbf{y}' \mid \mathbf{x}) - \log p(\mathbf{y}, \mathbf{y}') + \log q(\mathbf{y}, \mathbf{y}').$$

Alternatively, we can use $q(\mathbf{y}, \mathbf{y}')/p(\mathbf{y}, \mathbf{y}')$ to correct for target shift during training [Zhang et al., 2013].

## 3.2 Estimating $p(\mathbf{y}, \mathbf{y}')$

GMTL requires estimating $p(\mathbf{y}, \mathbf{y}')$, which is trivial for categorical targets when $\mathcal{Y} \times \mathcal{Y}'$ is small. The maximum likelihood estimate of $p(\mathbf{y}, \mathbf{y}')$ is obtained by counting the number of occurrences of the pair $(\mathbf{y}, \mathbf{y}')$, and dividing by the total across all pairs. However, there is an important precaution to be made, since it is possible that certain pairs of $(\mathbf{y}, \mathbf{y}')$ are never observed. This is problematic, since $p(\mathbf{y}, \mathbf{y}') = 0$ results in taking a logarithm of zero in Eq. 5. This can be addressed with additive smoothing, where a pseudocount $\epsilon > 0$ is added to the count for each pair prior to normalization. The issue of sparsity in the target distribution is the primary challenge in scaling GMTL to a larger number of tasks. Another complication is that in MTL, it is not unusual for a subset of targets to be missing for a given example. When this is the case, the counting approach only works with the subset of examples for which all targets are present, which can lead to poor estimates.

# 4    Toy problem

Before empirically validating GMTL on real datasets, we use synthetic data to demonstrate how DMTL and GMTL behave differently in the presence of target-causing confounding. We construct a data generating process (DGP) where a target-causing confounder induces a spurious dependency between a scalar input $X$ and binary target $Y$.

Suppose there are two targets $Y, Y' \in \{0, 1\}$ that are bivariate Bernoulli. This distribution can be specified in terms of $P(Y = 1) = \theta$, $P(Y' = 1) = \theta'$, and $\mathrm{Cov}[Y, Y']$. We keep $\theta$ and $\theta'$ fixed, and vary the value of $\mathrm{Cov}[Y, Y']$ in order to induce target shift. $\mathrm{Cov}[Y, Y']$ is a target-causing confounder in this context. $X$ is generated by a mixture of Gaussians with latent variable $Z = 2Y + Y'$, i.e. $X \mid z \sim N(z, \sigma_z^2)$. We fix $\theta = \theta' = 0.5$, $\sigma_0^2 = \sigma_1^2 = 0.4$, and $\sigma_2^2 = \sigma_3^2 = 0.6$, and vary $\mathrm{Cov}[Y, Y']$ from $-0.2$ to $0.2$. In other words, the only property of the DGP that we vary is $\mathrm{Cov}[Y, Y']$.

Fig. 2 shows how the decision boundaries of DMTL and GMTL with $\alpha = 1$ are affected by variation in the target-causing confounder $\mathrm{Cov}[Y, Y']$. The decision boundary of DMTL is the value of $X$ for which the solution to $\mathrm{argmax}_y \log P(y \mid x)$ changes, where $Y'$ has been marginalized out. In contrast, the decision boundary of GMTL is the value of $X$ where the $Y$ component of the solution to $\mathrm{argmax}_{y,y'} \log P(x \mid y, y')$ changes. The decision boundary for DMTL changes in response to $\mathrm{Cov}[Y, Y']$, while it remains constant for GMTL. This is because GMTL only uses the Gaussian densities for prediction, which have no connection to $\mathrm{Cov}[Y, Y']$.

In this example, we have shown that GMTL and DMTL arrive at different solutions by varying only the dependency between $Y$ and $Y'$ induced by the target-causing confounder $\mathrm{Cov}[Y, Y']$. The key difference between the two methods is that DMTL uses the spurious dependency between $X$ and $Y$, while GMTL does not. Which of the two methods achieves better predictive performance depends on the particular test distribution, since the spurious dependency will be predictive in ID settings, and not in OOD settings.

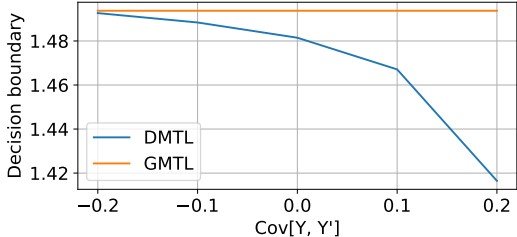

Figure 2: The decision boundary of DMTL changes w.r.t. $\mathrm{Cov}[Y, Y']$, while GMTL is invariant.

# 5    Experimental setup

We perform experiments to demonstrate that GMTL improves robustness to target shift. We evaluate across a wide range of target shifts ranging from mild to severe, and analyze how the optimal $\alpha$ changes w.r.t. the severity of target shift. Recall from Section 3.1 that we assume oracle access to the optimal $\alpha$, since we are focusing on validating GMTL, not on whether we can perform model selection w.r.t. unknown distributions, which is a separate open problem. The purpose of these experiments is to show that if we correctly assess how similar the training and test distributions are, we can benefit from using GMTL. In order to draw robust conclusions, we perform our experiments using two datasets, multiple pairs of classification tasks, and four MTL methods. Here, we describe our experimental setup.

## 5.1    Datasets

**Attributes of People**    The Attributes of People dataset [Bourdev et al., 2011] consists of 8,035 images of people. There are 4,013 examples in the training set, and 4,022 in the test set. Since the authors did not specify a validation set, we randomly sample 20% of the training set to use as the validation set. Each image comes labeled with up to nine binary attributes: male, long hair, glasses, hat, t-shirt, long sleeves, shorts, jeans, and long pants. We use each attribute as a binary classification

task, and predict whether the attribute appears in the image. We drop the gender attribute, since it has the potential for negative downstream applications [Wang and Kosinski, 2018]. We experiment with various pairs of tasks, taking turns specifying one as the main task, and the other as the auxiliary task. For each pair of tasks, we report the classification accuracy for the main task. Since not all attributes are labeled for each example, we consider the three pairs of tasks with the highest number of co-occurrences. These are: hat and long sleeves, long hair and hat, and glasses and hat.

**Taskonomy**   The Taskonomy dataset [Zamir et al., 2018] is of a much larger scale than the first dataset, containing approximately four million images of indoor scenes. We use a subset of the dataset that is provided by the authors for faster experimentation, with $548{,}785$ examples in the training set, $121{,}974$ in the validation set, and $90{,}658$ in the test set. Each image comes paired with 26 labels relevant to scene understanding, but since we are focusing on classification, we only use the object and scene annotations as targets. We use the object annotations as a 100-class object classification task, and the scene annotations as a 64-class scene classification task. Between object and scene classification, we take turns specifying one as the main task, and the other as the auxiliary task. We report the top-1 accuracy for the main task.

## 5.2   Models

**No Parameter Sharing (NPS)**   The first MTL method we use is a trivial combination of two single-task networks. That is, $p(\mathbf{y} \mid \mathbf{x})$ and $p(\mathbf{y}' \mid \mathbf{x})$ are trained separately. We call this No Parameter Sharing (NPS). This is equivalent to single-task learning during training, but becomes different during inference for $\alpha > 0$, since the single-task networks influence one another through the $-\alpha \log p(\mathbf{y}, \mathbf{y}')$ term. While NPS is primarily meant to be a simple setting for comparing GMTL and DMTL, it offers some practical utility as well. MTL methods are notoriously difficult to train, and it is often challenging to get them to perform better than the single-task baseline [Alonso and Plank, 2017]. NPS combined with GMTL offers a way to combine two single-task networks to reap the benefits of MTL, just by estimating $p(\mathbf{y}, \mathbf{y}')$ and predicting jointly over the targets.

**Shared Trunk Networks (STN)**   The second MTL method is a hard parameter sharing method that we call Shared Trunk Networks (STN), where all weights except for the final classification layer are shared. This is not an effective architecture for many of the tasks, since the predictive performance is worse than the single-task baseline. This is common for STN [Alonso and Plank, 2017], but we include it for completeness because it is the canonical MTL architecture. The results comparing single-task and MTL methods on the ID test set are in the supplementary material.

**Cross-stitch Networks (CSN)**   For the third MTL method, we use a soft parameter sharing method called Cross-stitch Networks (CSN) [Misra et al., 2016]. CSN takes two trained single-task networks, and takes a linear combination of the activations at each layer during the forward pass. Training CSN involves learning the coefficients of the linear combinations, as well as fine-tuning the single-task network weights. This network performs better than the single-task baseline across most datasets and tasks.

**Full Parameter Sharing (FPS)**   For our experiments on Attributes of People, we model $p(\mathbf{y}, \mathbf{y}' \mid \mathbf{x})$ directly as a four-class classification, without assuming factorization. Since this is a single network with all weights shared across both tasks, we call this Full Parameter Sharing (FPS). We include FPS to assess the appropriateness of letting $p(\mathbf{y}, \mathbf{y}' \mid \mathbf{x})$ factorize in the other MTL approaches, as we discussed in Section 3. We omit FPS on Taskonomy due to the complications of $\mathcal{Y} \times \mathcal{Y}'$ being large, and leave this to future work.

For all datasets and tasks, we use ResNet-50 [He et al., 2016] pretrained on ImageNet [Deng et al., 2009] for the single-task networks. We train using Adam [Kingma and Ba, 2015] with $L_2$ regularization, and tune the learning rate and regularization multiplier. For data augmentation, during training we resize the images to $256 \times 256$, randomly crop them to $224 \times 224$ and randomly horizontally flip them. During validation and testing, we resize the images to $256 \times 256$, and center crop them to $224 \times 224$. For all experiments, we train single-task networks from five random initializations. Details regarding other hyperparameters are included in the supplementary material.

### 5.3 Simulating target shift with importance sampling

In our experiments, we aim to evaluate across a wide range of target shifts. Since it is impractical to collect many different test sets, we simulate target shifts using importance sampling. Suppose the original test distribution is $p(\mathbf{x}, \mathbf{y}, \mathbf{y}') = p(\mathbf{x} \mid \mathbf{y}, \mathbf{y}')p(\mathbf{y}, \mathbf{y}')$, and the simulated test distribution is $q(\mathbf{x}, \mathbf{y}, \mathbf{y}') = p(\mathbf{x} \mid \mathbf{y}, \mathbf{y}')q(\mathbf{y}, \mathbf{y}')$. $p(\mathbf{x} \mid \mathbf{y}, \mathbf{y}')$ is invariant due to our assumption that it is a causal relation. Therefore, the expected accuracy $l(\mathbf{x}, \mathbf{y}, \mathbf{y}')$ under the simulated test distribution $q(\mathbf{x}, \mathbf{y}, \mathbf{y}')$ is given by

$$
\begin{aligned}
\mathbb{E}_{q(\mathbf{x},\mathbf{y},\mathbf{y}')}[l(\mathbf{x}, \mathbf{y}, \mathbf{y}')] &= \mathbb{E}_{p(\mathbf{x}|\mathbf{y},\mathbf{y}')q(\mathbf{y},\mathbf{y}')}[l(\mathbf{x}, \mathbf{y}, \mathbf{y}')] \\
&= \mathbb{E}_{p(\mathbf{x},\mathbf{y},\mathbf{y}')}\left[\frac{q(\mathbf{y},\mathbf{y}')}{p(\mathbf{y},\mathbf{y}')}l(\mathbf{x}, \mathbf{y}, \mathbf{y}')\right] \\
&\approx \frac{1}{N}\sum_{n=1}^{N}\frac{q(\mathbf{y}^{(n)},\mathbf{y}'^{(n)})}{p(\mathbf{y}^{(n)},\mathbf{y}'^{(n)})}l(\mathbf{x}^{(n)}, \mathbf{y}^{(n)}, \mathbf{y}'^{(n)}) \\
&\approx \sum_{n=1}^{N}\frac{q(\mathbf{y}^{(n)},\mathbf{y}'^{(n)})}{p(\mathbf{y}^{(n)},\mathbf{y}'^{(n)})}\Bigg/ \sum_{m=1}^{N}\frac{q(\mathbf{y}^{(m)},\mathbf{y}'^{(m)})}{p(\mathbf{y}^{(m)},\mathbf{y}'^{(m)})}l(\mathbf{x}^{(n)}, \mathbf{y}^{(n)}, \mathbf{y}'^{(n)}).
\end{aligned}
$$

Now that we can evaluate across a wide range of OOD $q(\mathbf{y}, \mathbf{y}')$'s, we need a way to quantify the severity of the target shift between $p(\mathbf{y}, \mathbf{y}')$ and $q(\mathbf{y}, \mathbf{y}')$.

### 5.4 Measuring the severity of target shift between $p(\mathbf{y}, \mathbf{y}')$ and $q(\mathbf{y}, \mathbf{y}')$

Our metric to measure the severity of target shift between $p(\mathbf{y}, \mathbf{y}')$ and $q(\mathbf{y}, \mathbf{y}')$ must be such that predictive performance degrades w.r.t. the severity of shift. Not all metrics satisfy this criteria. To build intuition, consider a Bernoulli distribution where $P(Y = 1) = \theta$. If we use a norm-based metric such as total variation, this treats a change in $\theta$ from 0.6 to 0.8 and a change from 0.6 to 0.4 as being the same. However, from the point of view of prediction, the latter should be more detrimental, since it reverses the ranking of the classes.

Similarly, Kullback-Leibler (KL) divergence is a common choice for measuring the distance between distributions, but it suffers from a related problem. Consider $\theta$ from above changing from 0.9 to 0.9999, and from 0.9 to 0.4. Due to taking the logarithm of a small number, KL divergence considers the first to be a more severe shift. This is undesirable, because the change from 0.9 to 0.4 represents a reversal in the ranking of the classes.

These examples point to a need for a ranking-based metric, since a significant change to the ranking of classes is detrimental to predictive performance. Weighted rank correlation satisfies this. If the weighted rank correlation between $p(\mathbf{y}, \mathbf{y}')$ and $q(\mathbf{y}, \mathbf{y}')$ is positive with large magnitude, it means that there are no significant changes in the ranking of classes, and that the target shift is not severe. In contrast, if the weighted rank correlation is negative with large magnitude, it implies the rankings of classes has changed significantly. This constitutes a more severe target shift. We therefore use weighted Kendall's $\tau$ [Shieh, 1998], which we henceforth refer to as $\tau$, to measure the severity of target shift between $p(\mathbf{y}, \mathbf{y}')$ and $q(\mathbf{y}, \mathbf{y}')$. We provide the definition of $\tau$ in the supplementary material. It relies on a weighting function, for which we use $1/(r + 1)$, where $r$ is the ranking.

### 5.5 Sampling $q(\mathbf{y}, \mathbf{y}')$

Our choice of distance metric informs how to sample $q(\mathbf{y}, \mathbf{y}')$, since our goal is to evaluate across a wide range of target shift severities. We designed a method which shuffles $\log p(\mathbf{y}, \mathbf{y}')$ and perturbs it with noise. To be precise, let $f(\mathbf{y}, \mathbf{y}') \in \{1, 2, \ldots, |\mathcal{Y} \times \mathcal{Y}'|\}$ such that each $(\mathbf{y}, \mathbf{y}')$ is assigned a unique integer index. We use $p_{f(\mathbf{y},\mathbf{y}')} = p(\mathbf{y}, \mathbf{y}')$ Let $i = \lambda \cdot |\mathcal{Y} \times \mathcal{Y}'|$, where $\lambda \sim \mathcal{U}(0, 0.5)$. We randomly shuffle these indices with $\pi : \{1, 2, \ldots, i\} \to \{1, 2, \ldots, \lambda \times i\}$. Then, we define $q(\mathbf{y}, \mathbf{y}')$ by

$$
q(\mathbf{y}, \mathbf{y}') \propto \begin{cases} \exp(\log p(\mathbf{y}, \mathbf{y}') + \epsilon) & \text{if } f(\mathbf{y}, \mathbf{y}') > i \\ \exp(\log p_{\pi(f(\mathbf{y},\mathbf{y}'))} + \epsilon) & \text{if } f(\mathbf{y}, \mathbf{y}') \leq i, \end{cases}
$$

where $\epsilon \sim \mathcal{N}(0, \sigma^2)$, and $\sigma \sim \mathcal{U}(10^{-12}, 5)$.

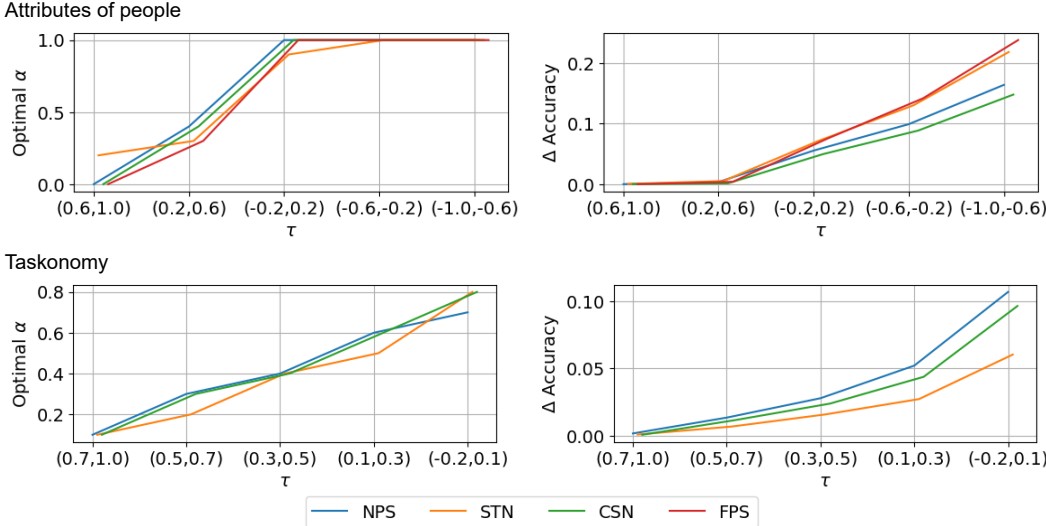

Figure 3: In order to evaluate whether GMTL improves robustness to target shift, we estimate OOD accuracy across a range of target shift severities and a range of $\alpha \in [0, 1]$. The results are grouped w.r.t. the severity of target shift $\tau$, and we report the optimal $\alpha$ for each group. As seen in the left subfigures, as the target shift becomes more severe from left to right, the optimal $\alpha$ monotonically increases for both Attributes of People (top left) and Taskonomy (bottom left). This is because a larger $\alpha$ removes more spurious dependencies induced by target-causing confounding, which is more beneficial when the severity of shift increases. In the right subfigures, the vertical axis is the difference in accuracy between GMTL and DMTL. The improvement in accuracy from using GMTL increases w.r.t. the severity of target shift for both Attributes of People (top right) and Taskonomy (bottom right). The reason is that the spurious dependencies induced by target-causing confounding become less predictive as the severity of shift increases, and therefore removing them yields larger improvements in accuracy.

## 6   Results

We compute the OOD test set accuracy across target shifts ranging from mild to severe, where $p(\mathbf{y}, \mathbf{y}')$ is the original test set distribution, and $q(\mathbf{y}, \mathbf{y}')$ is a randomly sampled OOD distribution. First, we sample an OOD $q(\mathbf{y}, \mathbf{y}')$, and compute $\tau$ relative to $p(\mathbf{y}, \mathbf{y}')$. $\tau$ represents the severity of target shift between $q(\mathbf{y}, \mathbf{y}')$ and the original test distribution $p(\mathbf{y}, \mathbf{y}')$. We then use importance sampling to compute the OOD accuracy under each $q(\mathbf{y}, \mathbf{y}')$ across a range of $\alpha \in [0, 1]$.

The results are split into five groups that are equally spaced in terms of $\tau$. Each group represents the accuracies for a range of $\alpha$ under a particular severity of target shift. For each group, we compute the optimal value of $\alpha$ that attains the highest accuracy. We also compute the difference in accuracy between GMTL and DMTL at the optimal value of $\alpha$ for each group.

We observe strikingly similar results for both datasets and all pairs of tasks. As the target shift becomes more severe, the optimal $\alpha$ increases. A representative example for each dataset is shown in the left subfigures in Fig. 3, and the rest are in the supplementary material. This observation matches our understanding of $\alpha$ from Section 3.1. Since $\alpha$ is our degree of belief on how similar $p(\mathbf{y}, \mathbf{y}')$ and $q(\mathbf{y}, \mathbf{y}')$ are, the optimal $\alpha$ tends to be higher when the target shift is more severe. Increasing $\alpha$ removes more of the spurious dependencies induced by target-causing confounding, which is more beneficial for severe target shifts.

Also, as the severity of target shift increases, the difference in accuracy between GMTL and DMTL for the optimal $\alpha$ increases as well. This can be seen in the right subfigures in Fig. 3, and is also in-line with our expectations. As the target shift becomes more severe, the spurious dependencies induced by target-causing confounding become less predictive, and removing them yields larger improvements in predictive performance.

Both patterns hold very similarly for all MTL methods. This suggests that it is our formulation of GMTL that is significant, rather than the particular parameterization used to learn $p(\mathbf{y}, \mathbf{y}' \mid \mathbf{x})$. Both the optimal $\alpha$, as well as the corresponding improvement in accuracy using GMTL, increase monotonically w.r.t. the severity of target shift. This is a strong relation that holds across datasets, pairs of tasks, and MTL methods. To quantify the generality of this result, we aggregate all of our results, and compute the correlation between $\tau$ and the optimal $\alpha$. Since $\tau$ is used in the context of an interval, we take the midpoint of each interval. This results in a correlation of $-0.847$. This indicates a strong positive correlation between the severity of target shift and the optimal $\alpha$. The generality of this relation is strong evidence that GMTL improves robustness to target-causing confounding, and also that our interpretation of $\alpha$ is correct. Our experimental results using the $\alpha$-selection heuristic described in Section 3.1, as well as other results showing the accuracy for a range of $\alpha$, not just the optimal one, are in the supplementary material.

# 7 Conclusion

We presented generative multitask learning (GMTL), an approach for causal ML in the multitask setting that only changes the inference objective of conventional MTL. Our approach mitigates the effect of target-causing confounders, which are variables that cause the targets, but not the input. This removes spurious dependencies between the input and targets, and improves robustness to target shift. Looking forward, we are excited by the new perspectives that causality can bring to improve weaknesses in ML. Our work demonstrates the potential for this in the context of MTL. We plan to investigate what other weaknesses in ML can be improved by adopting a causal perspective.

# 8 Acknowledgements

This work was supported by grants from the National Institutes of Health (P41EB017183), the National Science Foundation (HDR-1922658), the Gordon and Betty Moore Foundation (9683), and Samsung Advanced Institute of Technology (Next Generation Deep Learning: From Pattern Recognition to AI).

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
