# A  Appendix

## A.1  Definitions

Weighted Kendall's $\tau$ is a measure of rank correlation between two vectors $\mathbf{x}$ and $\mathbf{y}$, both with length $N$. Using the authors' notation, let $(i, R_i)$ for $i = 1, \ldots, N$ be pairs such that $R_i$ is the rank of the element in $\mathbf{y}$ whose corresponding $\mathbf{x}$ value has rank $i$. Let $w(i, j)$ be a bounded and symmetric weight function that maps to $\mathbb{R}$, and denote its value at $(i, j)$ as $w_{ij}$. Weighted Kendall's $\tau$ is defined as

$$\tau = \frac{\sum_{i \neq j} w_{ij} \operatorname{sgn}(i - j) \operatorname{sgn}(R_i - R_j)}{\sum_{i,j} w_{ij} - \sum_i w_{ii}},$$

where

$$\operatorname{sgn}(x) = \begin{cases} -1 & \text{if } x < 0 \\ 0 & \text{if } x = 0 \\ 1 & \text{if } x > 0. \end{cases}$$

## A.2  Derivations

GMTL corresponds to replacing the empirical target distribution $p(\mathbf{y}, \mathbf{y}')$ with a distribution that is proportional to $p(\mathbf{y}, \mathbf{y}')^{1-\alpha}$. To see why, suppose the empirical joint distribution is $p(\mathbf{x}, \mathbf{y}, \mathbf{y}')$ and we undergo a target shift such that the test distribution is $q(\mathbf{x}, \mathbf{y}, \mathbf{y}') = p(\mathbf{x} \mid \mathbf{y}, \mathbf{y}')q(\mathbf{y}, \mathbf{y}')$. If we assume that $p(\mathbf{y}, \mathbf{y}' \mid x)$ factorizes, then predicting optimally w.r.t. $q(\mathbf{x}, \mathbf{y}, \mathbf{y}')$ gives us

$$
\begin{aligned}
\operatorname*{argmax}_{\mathbf{y}, \mathbf{y}'} \log q(\mathbf{x}, \mathbf{y}, \mathbf{y}') &= \operatorname*{argmax}_{\mathbf{y}, \mathbf{y}'} \log p(\mathbf{x} \mid \mathbf{y}, \mathbf{y}') + \log q(\mathbf{y}, \mathbf{y}') \\
&= \operatorname*{argmax}_{\mathbf{y}, \mathbf{y}'} \log p(\mathbf{x} \mid \mathbf{y}, \mathbf{y}') + \log q(\mathbf{y}, \mathbf{y}') \\
&= \operatorname*{argmax}_{\mathbf{y}, \mathbf{y}'} \log p(\mathbf{y}, \mathbf{y}' \mid x) - \log p(\mathbf{y}, \mathbf{y}') + \log q(\mathbf{y}, \mathbf{y}') \\
&= \operatorname*{argmax}_{\mathbf{y}, \mathbf{y}'} \log p(\mathbf{y} \mid \mathbf{x}) + \log p(\mathbf{y}' \mid \mathbf{x}) \underbrace{- \log p(\mathbf{y}, \mathbf{y}') + \log q(\mathbf{y}, \mathbf{y}')}_{-\alpha \log p(\mathbf{y}, \mathbf{y}')}.
\end{aligned}
\tag{1}
$$

Notice that Eq. 1 resembles GMTL. If we write

$$
\begin{aligned}
-\log p(\mathbf{y}, \mathbf{y}') + \log q(\mathbf{y}, \mathbf{y}') &\propto -\alpha \log p(\mathbf{y}, \mathbf{y}') \\
\log q(\mathbf{y}, \mathbf{y}') &\propto (1 - \alpha) \log p(\mathbf{y}, \mathbf{y}'),
\end{aligned}
$$

this shows that GMTL corresponds to assuming $q(\mathbf{y}, \mathbf{y}') \propto p(\mathbf{y}, \mathbf{y}')^{1-\alpha}$.

## A.3  Reproducibility

The code required to fully reproduce our experiments, including the configuration files that contain all hyperparameter values, is available at `https://github.com/nyukat/generative-multitask-learning`. We ran our experiments on a single NVIDIA RTX8000 GPU on our high performance computing system. For convenience, here are the three functions relevant to the GMTL inference objective. When executed sequentially, they take as input the task-specific log probabilities $\log p(\mathbf{y} \mid \mathbf{x})$ and $\log p(\mathbf{y}' \mid \mathbf{x})$, the target distribution $p(\mathbf{y}, \mathbf{y}')$, and the parameter $\alpha$, and returns

$$\operatorname*{argmax}_{\mathbf{y}, \mathbf{y}'} \log p(\mathbf{y} \mid \mathbf{x}) + \log p(\mathbf{y}' \mid \mathbf{x}) - \alpha \log p(\mathbf{y}, \mathbf{y}').$$

```python
def to_log_joint_pred(log_marginals):
    '''
    Input: [log p(y | x), log p(y' | x)]
```

```python
    Output: log p(y, y' | x) = log p(y | x) + log p(y' | x)
    '''
    log_joint_shape = [elem.shape[1] for elem in log_marginals]
    n_examples = len(log_marginals[0])
    log_joint = np.full(log_joint_shape + [n_examples], np.nan)
    for flat_idx in range(np.prod(log_joint_shape)):
        unflat_idx = np.unravel_index(flat_idx, log_joint_shape)
        log_prob = 0
        for task_idx, class_idx in enumerate(unflat_idx):
            log_prob += log_marginals[task_idx][:, class_idx]
        log_joint[unflat_idx] = log_prob
    log_joint = np.moveaxis(log_joint, -1, 0)
    return log_joint

def to_generative_pred(log_joint, alpha, log_prior):
    '''
    Input: log p(y, y' | x)
    Output: log p(y, y' | x) - alpha * log p(y, y')
    '''
    return log_joint - alpha * log_prior

def to_class_pred(log_joint):
    '''
    Input: log p(y, y' | x) - alpha * log p(y, y')
    Output: argmax_{y, y'} log p(y, y' | x) - alpha * log p(y, y')
    '''
    class_pred = []
    log_joint_shape = log_joint.shape[1:]
    for pred_elem in log_joint:
        class_pred.append(np.unravel_index(np.argmax(pred_elem), log_joint_shape))
    class_pred = np.array(class_pred)
    return class_pred
```

## A.4   In-distribution test set accuracy

We report the in-distribution (ID) test set accuracy for each task and MTL method. These results use the test set that originally came with each dataset. The purpose of these results is to show how the MTL methods compare to one another in the ID setting. Since we know that the training and test distributions are similar, we set $\alpha = 0$, in which case NPS is equivalent to single-task learning (STL). STL is a strong baseline, but CSN performs better than it for the majority of tasks. The performance of STN is mixed, which is consistent with the MTL literature. STN does significantly worse than the STL baseline on Taskonomy despite extensive hyperparameter tuning, but we include the results because it is the canonical MTL architecture.

Table 1: Test set accuracy for hat and long sleeves on Attributes of People.

|     | Hat | Long sleeves |
| --- | --- | --- |
| NPS | $0.917 \pm 0.003$ | $0.853 \pm 0.004$ |
| STN | $0.912 \pm 0.002$ | $0.852 \pm 0.003$ |
| CSN | $0.920 \pm 0.001$ | $0.861 \pm 0.005$ |

Table 2: Test set accuracy for long hair and hat on Attributes of People.

|      | Long hair         | Hat               |
| ---- | ----------------- | ----------------- |
| NPS  | $0.858 \pm 0.003$ | $0.917 \pm 0.003$ |
| STN  | $0.859 \pm 0.003$ | $0.916 \pm 0.000$ |
| CSN  | $0.860 \pm 0.002$ | $0.917 \pm 0.002$ |

Table 3: Test set accuracy for glasses and hat on Attributes of People.

|      | Glasses           | Hat               |
| ---- | ----------------- | ----------------- |
| NPS  | $0.843 \pm 0.011$ | $0.917 \pm 0.003$ |
| STN  | $0.855 \pm 0.003$ | $0.913 \pm 0.002$ |
| CSN  | $0.848 \pm 0.001$ | $0.917 \pm 0.002$ |

Table 4: Top-1 test set accuracy for object and scene classification on Taskonomy.

|      | Object            | Scene             |
| ---- | ----------------- | ----------------- |
| NPS  | $0.749 \pm 0.001$ | $0.730 \pm 0.001$ |
| STN  | $0.710 \pm 0.001$ | $0.719 \pm 0.001$ |
| CSN  | $0.750 \pm 0.001$ | $0.737 \pm 0.001$ |

## A.5   Anomalous results

As we will see throughout Sections A.6–A.8, the results for Attributes of People with long sleeves as the main task, and hat as the auxiliary task are anomalous. This is because our method for simulating and measuring target shift is not effective for this task. That is, the predictive performance increases w.r.t. the severity of target shift. The problem is that for this task, reversing the roles of the least and most common classes improves predictive performance. This runs counter to our intuition, as well as all other tasks in our experiments. Nonetheless, we include these results for completeness.

## A.6   Results using a heuristic to select $\alpha$

We report the results from using the $\alpha$-selection heuristic discussed in Section 3.2 of the main text. The heuristic is to choose the maximum possible $\alpha$ within an allowable budget of lost accuracy in the ID setting. For a given accuracy budget, we compute the validation set accuracy for a range of $\alpha$, and pick the largest $\alpha$ such that the accuracy is within the budget relative to $\alpha = 0$. Then, using the selected $\alpha$, we report the test set accuracy across a wide range of target shifts. The horizontal axis in these plots is the severity of target shift, increasing from left to right.

In each set of six figures below, we consider a pair of tasks. One of these tasks is designated the main task, and the other is the auxiliary task. This distinction is primarily important during training, when deciding which task loss to use for early stopping. We then report the accuracy only for the main task (left subfigures). We then reverse the roles of the main and auxiliary tasks for the right subfigures.

For all tasks except for the one mentioned in Section A.5, there is a very clear pattern that holds across the three MTL methods. Paying a small penalty in ID accuracy yields significant improvements in robustness to target shift. In some cases the improvement is very large, with roughly a 20% improvement for the most severe target shift (the left subfigures in Fig. 2). These results show that even a simple heuristic such as this can be very effective, which makes GMTL practical.

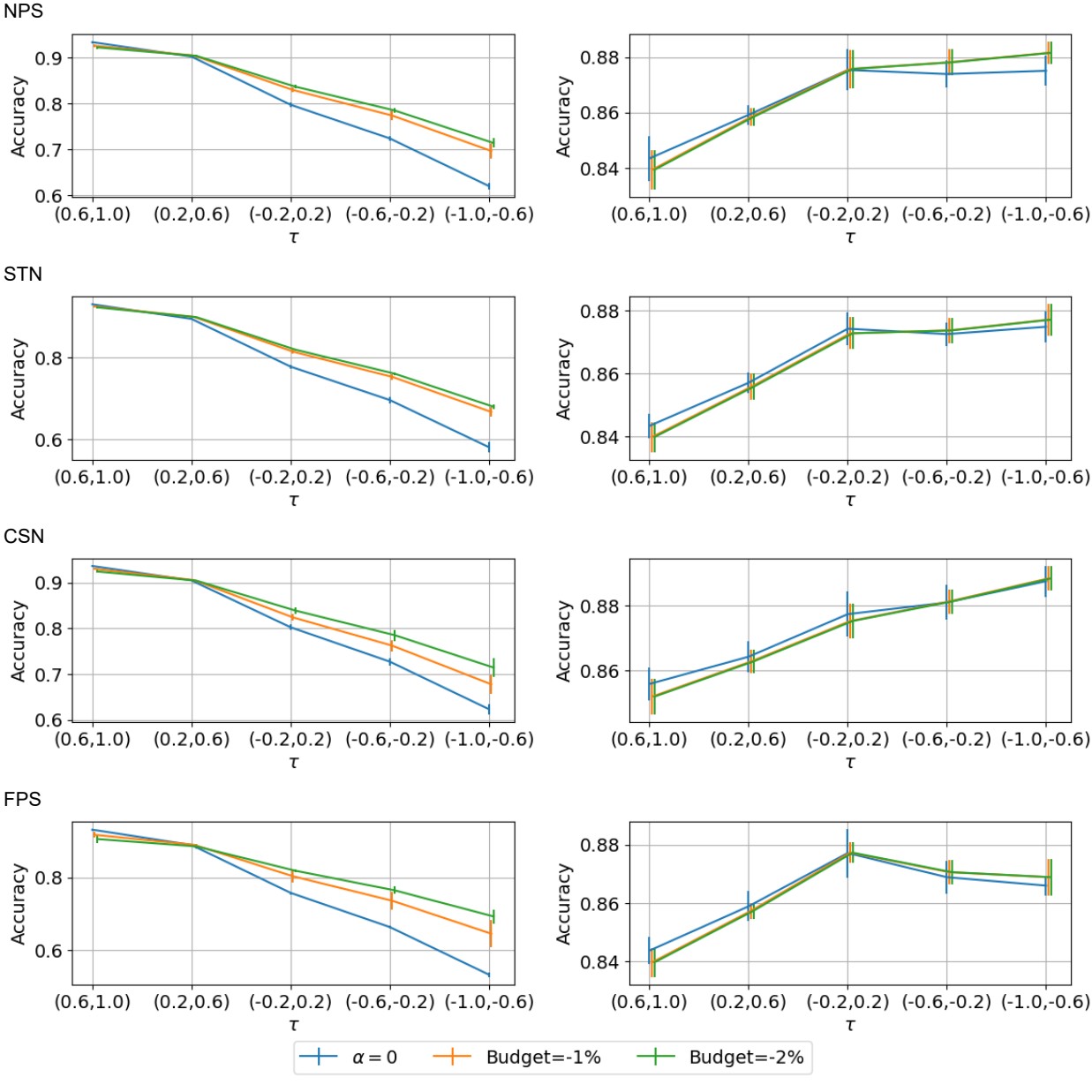

Figure 1: Attributes of people with hat as the main task, and long sleeves as the auxiliary task (left). The main and auxiliary tasks are exchanged in the right subfigures.

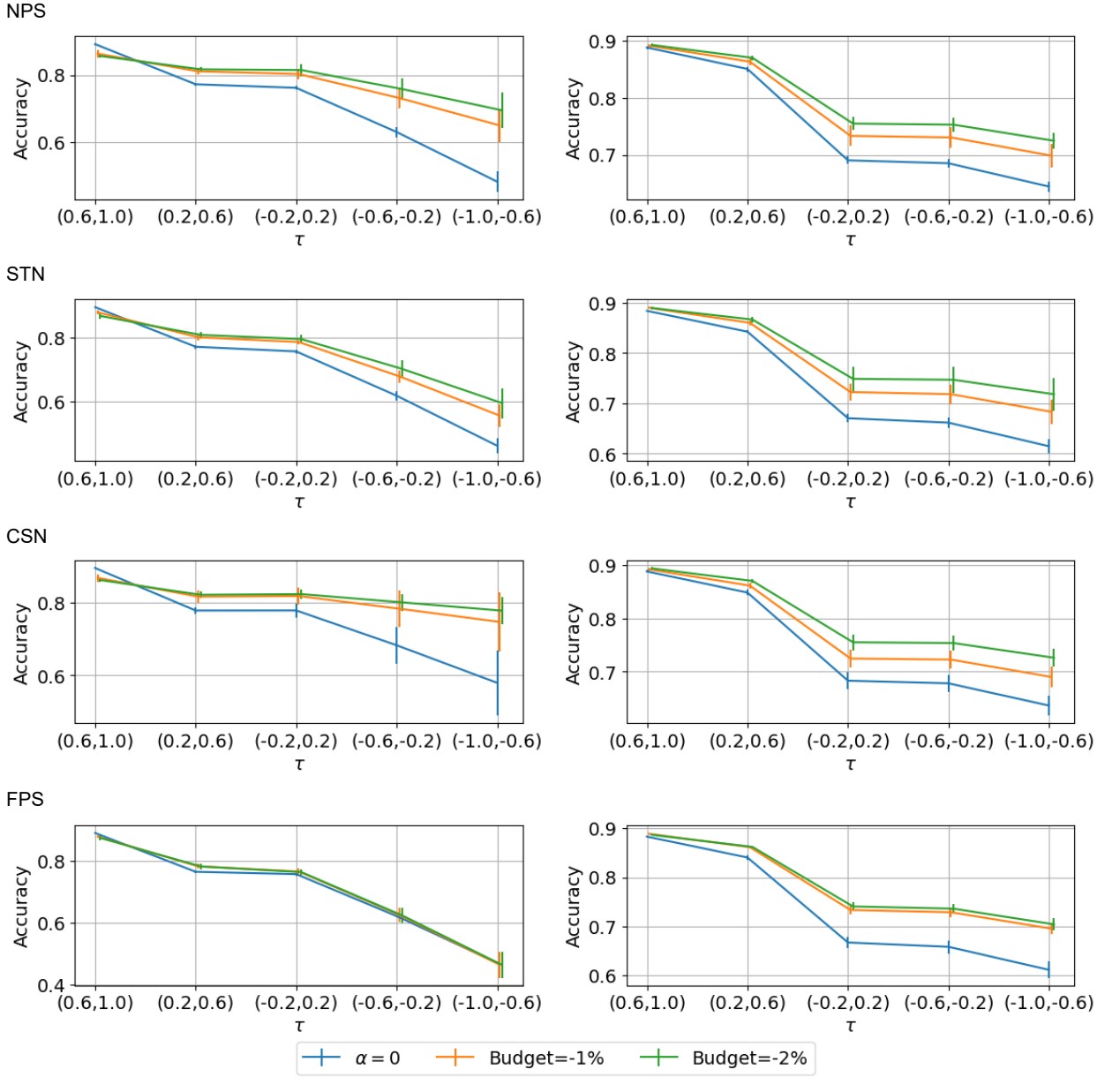

Figure 2: Attributes of people with long hair as the main task, and hat as the auxiliary task (left). The main and auxiliary tasks are exchanged in the right subfigures.

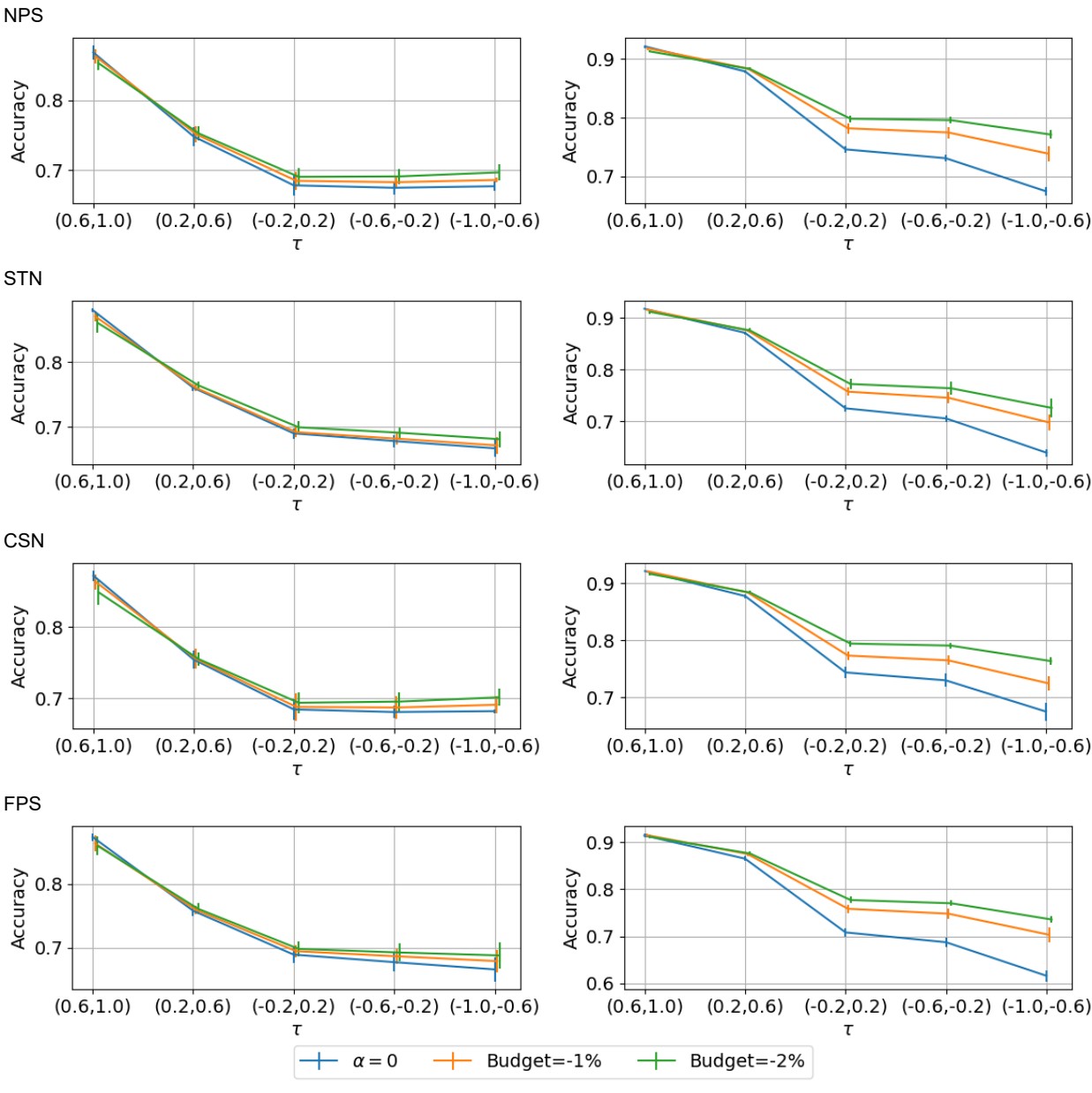

Figure 3: Attributes of people with glasses as the main task, and hat as the auxiliary task (left). The main and auxiliary tasks are exchanged in the right subfigures.

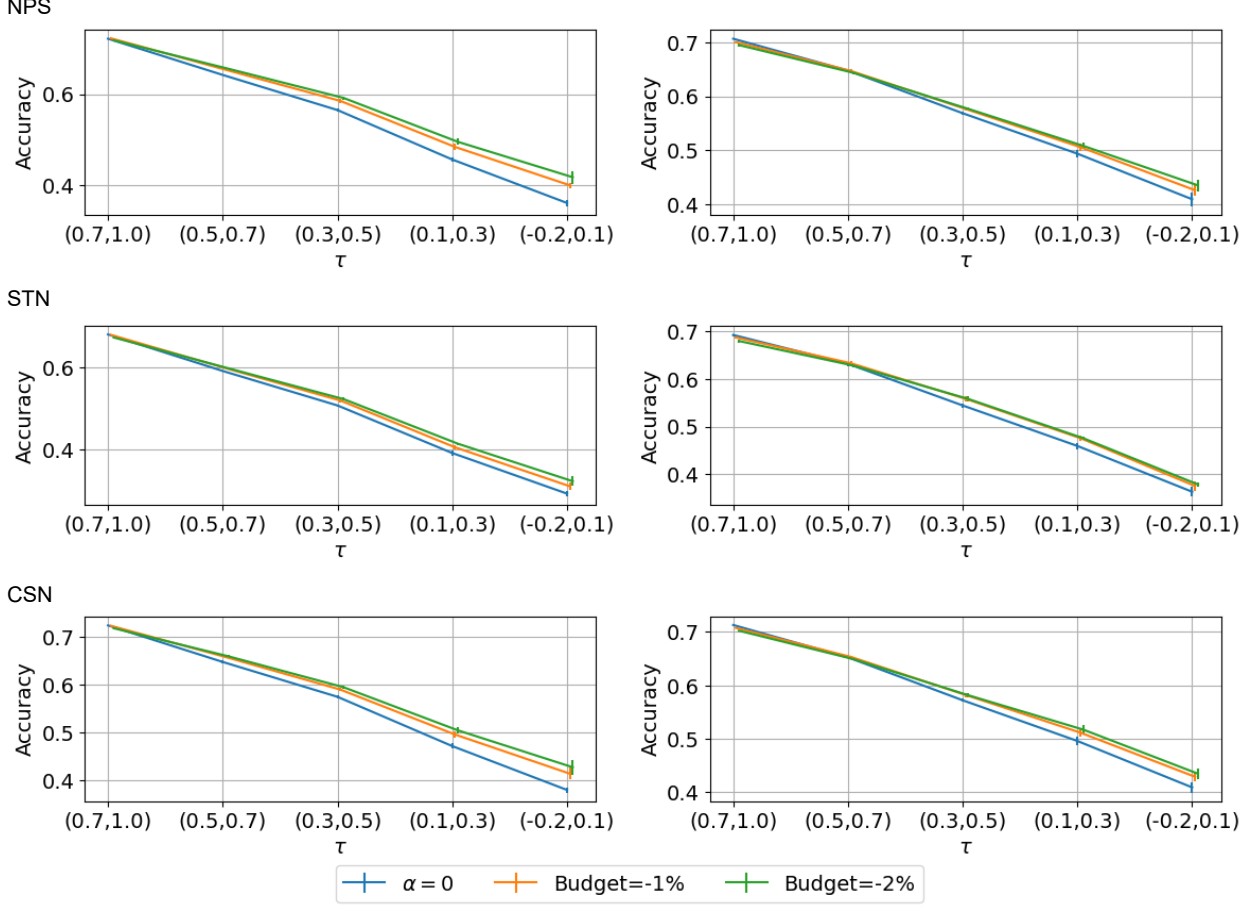

Figure 4: Taskonomy with object classification as the main task, and scene classification as the auxiliary task (left). The main and auxiliary tasks are exchanged in the right subfigures.

## A.7 Results with access to the optimal $\alpha$

These are the same type of results shown in Section 6 of the main text, but for the remaining tasks across both datasets. For all tasks except the one mentioned in Section A.5, the optimal alpha increases w.r.t. the severity of target shift, and there is a significant improvement in accuracy using the optimal alpha. These are the same observations that we made in the main text.

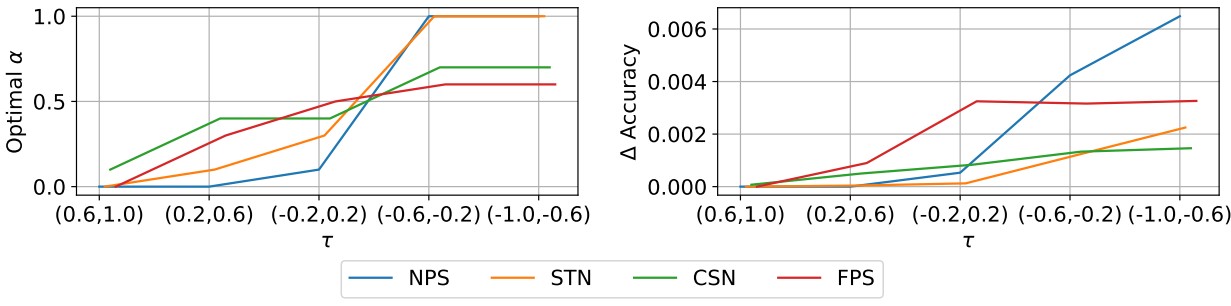

Figure 5: Attributes of people with long sleeves as the main task, and hat as the auxiliary task.

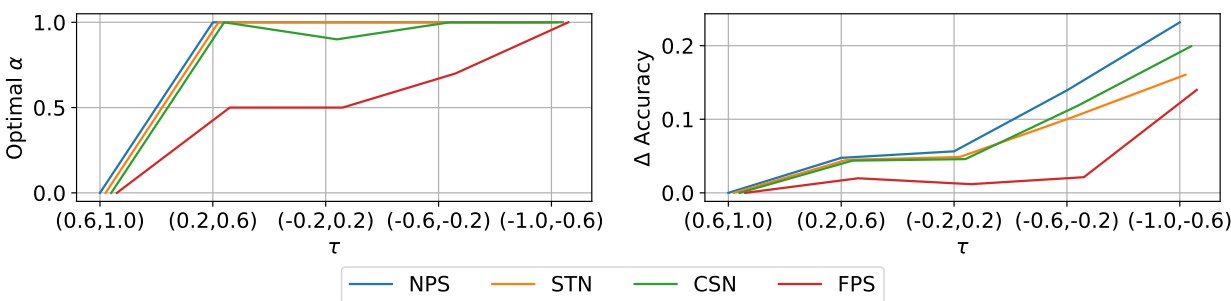

Figure 6: Attributes of people with long hair as the main task, and hat as the auxiliary task.

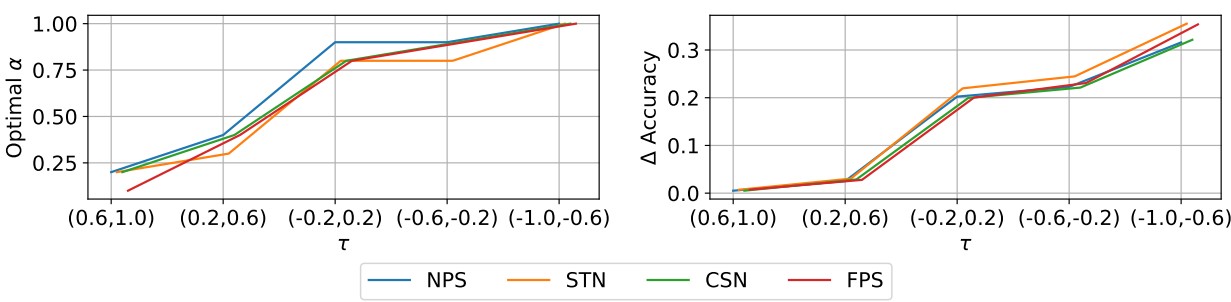

Figure 7: Attributes of people with hat as the main task, and long hair as the auxiliary task.

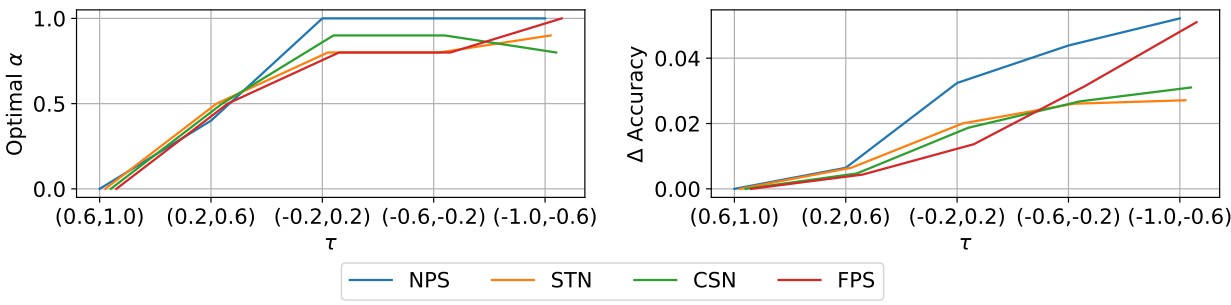

Figure 8: Attributes of people with glasses as the main task, and hat as the auxiliary task.

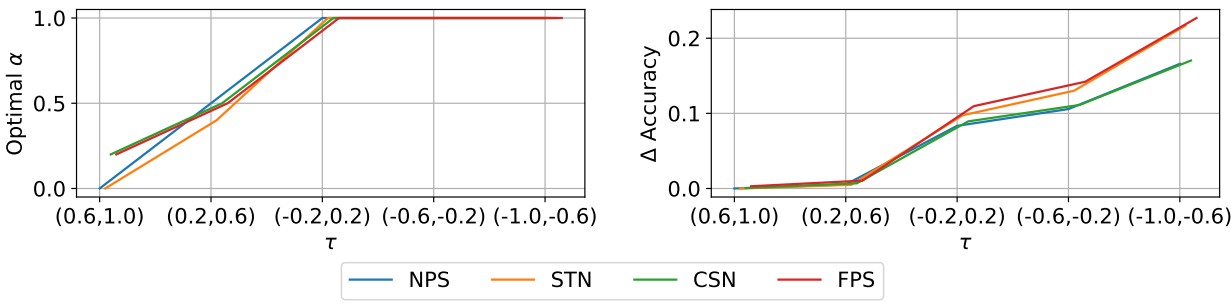

Figure 9: Attributes of people with hat as the main task, and glasses as the auxiliary task.

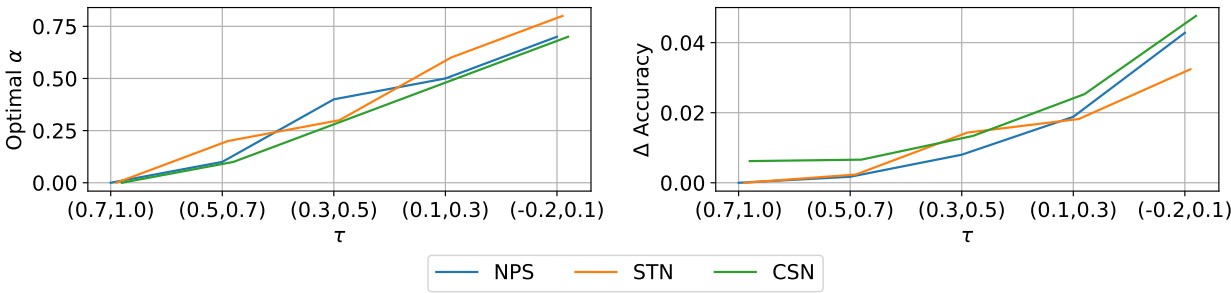

Figure 10: Taskonomy with scene classification as the main task, and object classification as the auxiliary task.

## A.8   Results for a range of $\alpha$

These results show the test accuracy for a range of $\alpha$, not just the optimal one. They can therefore be seen as a more granular view of the information presented in the left subfigures of Section A.7. In each set of four figures, each subfigure represents a different severity of target shift. Recall that the target shift is mildest when $\tau = 1$, and most severe when $\tau = -1$. These results support our interpretation of $\alpha$ as being a trade-off. That is, increasing $\alpha$ removes spurious dependencies that are predictive in the ID setting, and not in the out-of-distribution setting. Therefore, the optimal $\alpha$ tends to be small when the target shift is mild (top left), and large when the target shift is severe (bottom right). The optimal $\alpha$ tends to be somewhere in the middle when the target shift is in between being mild and severe. These conclusions hold consistently across all tasks, except for the one mentioned in Section A.5.

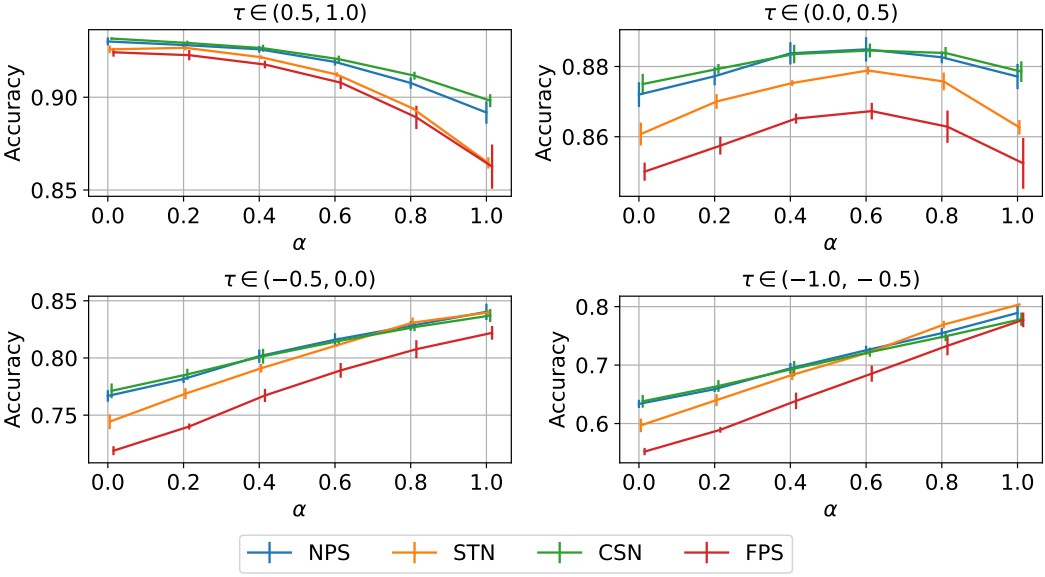

Figure 11: Attributes of people with hat as the main task, and long sleeves as the auxiliary task.

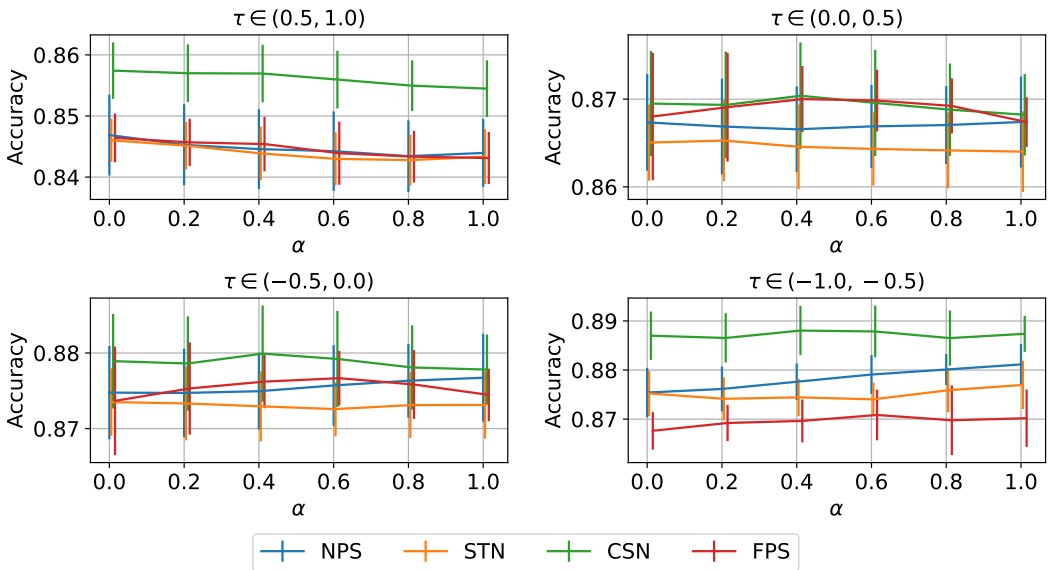

Figure 12: Attributes of people with long sleeves as the main task, and hat as the auxiliary task.

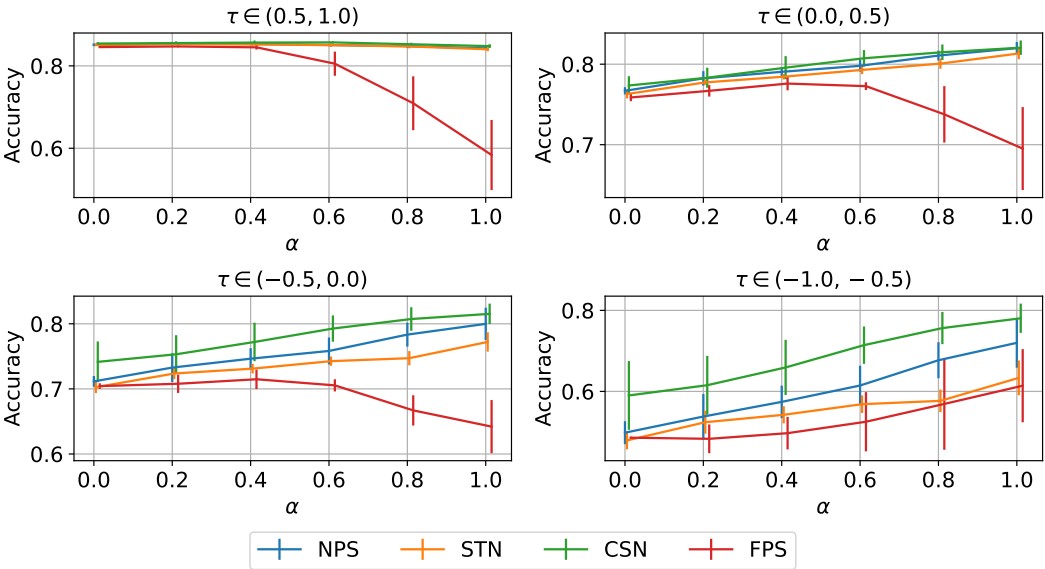

Figure 13: Attributes of people with long hair as the main task, and hat as the auxiliary task.

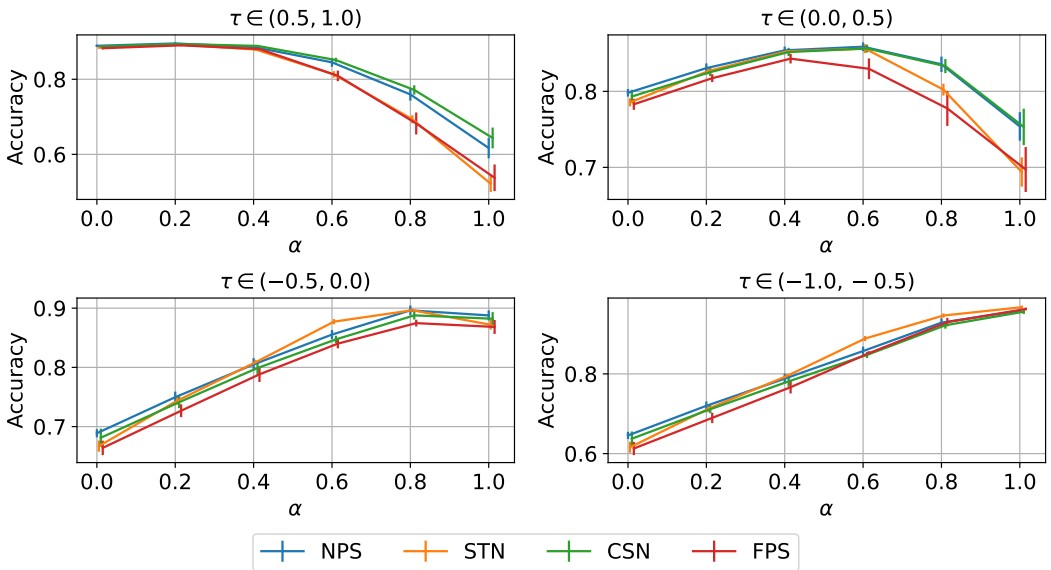

Figure 14: Attributes of people with hat as the main task, and long hair as the auxiliary task.

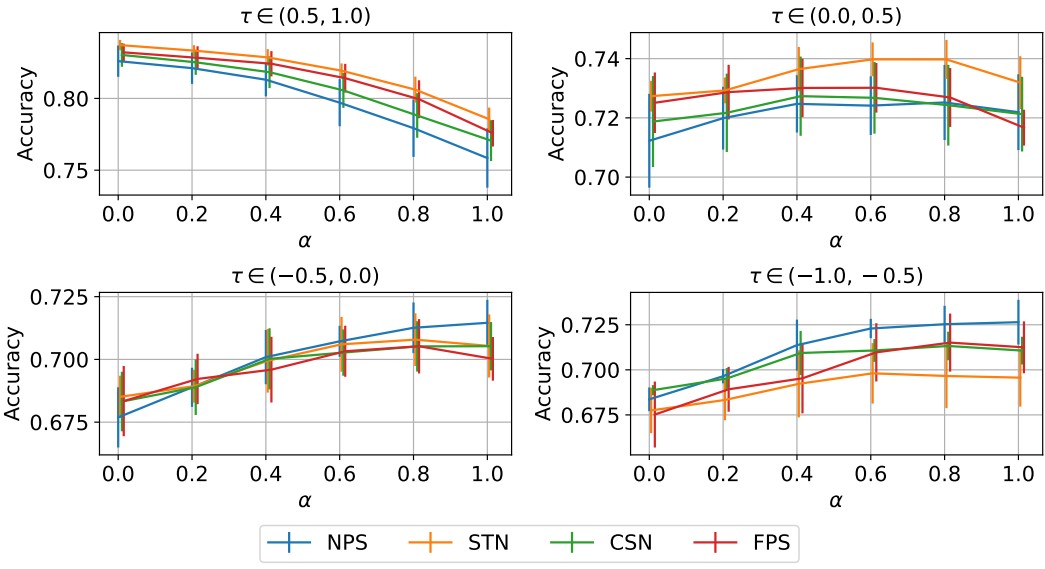

Figure 15: Attributes of people with glasses as the main task, and hat as the auxiliary task.

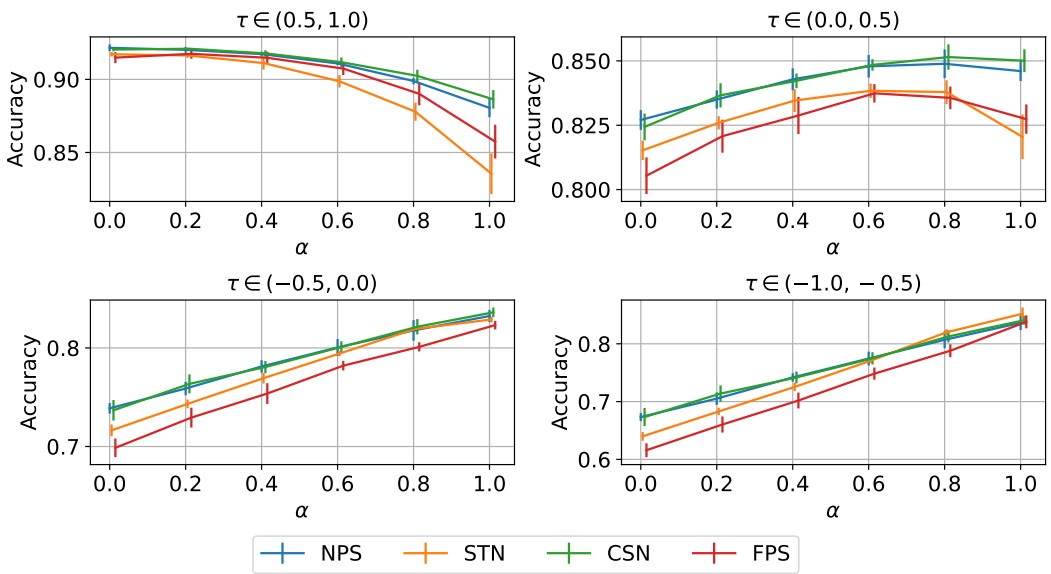

Figure 16: Attributes of people with hat as the main task, and glasses as the auxiliary task.

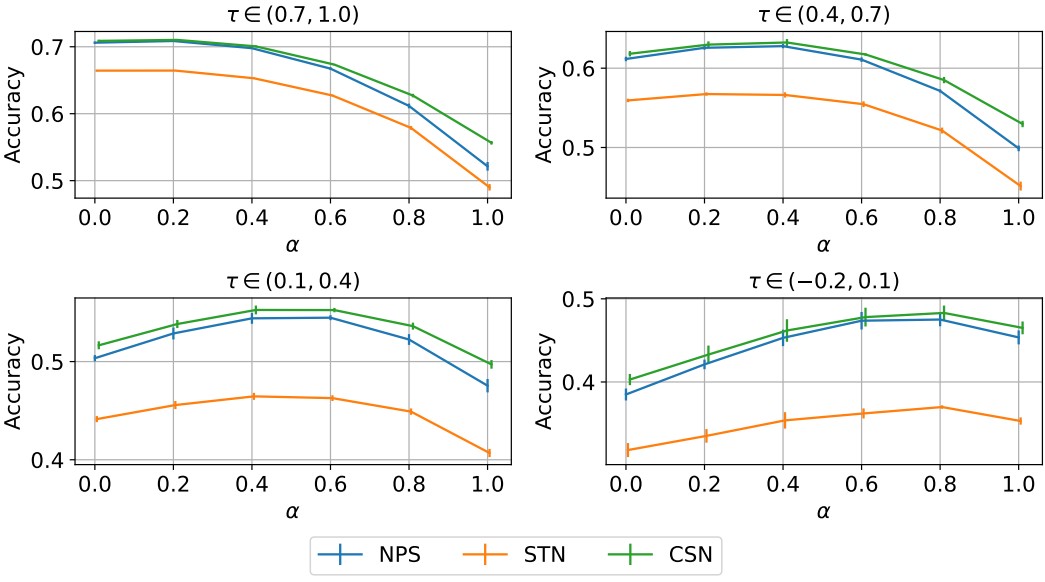

Figure 17: Taskonomy with object classification as the main task, and scene classification as the auxiliary task.

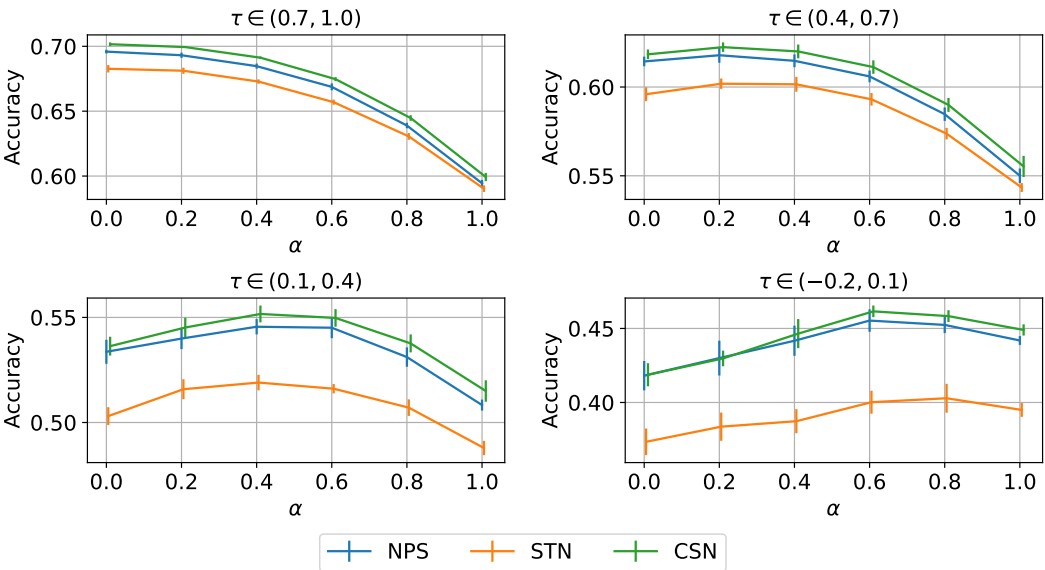

Figure 18: Taskonomy with scene classification as the main task, and object classification as the auxiliary task.