# OpenReview forum: "Generative multitask learning mitigates target-causing confounding"
_NeurIPS.cc/2022/Conference — NeurIPS 2022 Accept_

### Official Review · Reviewer_yfCS · 2022-07-09

**Rating:** 7
**Confidence:** 4
**Soundness:** 4 excellent
**Presentation:** 4 excellent
**Contribution:** 3 good

**Summary:**

This paper considers a more relaxed assumption in multitask learning, compared to previous works: targets are not conditionally independent given the input, but rather caused by a hidden confounding factor. Therefore, the authors propose generative multitask learning (GMTL), where the confounding factor is addressed in loss function. The GMTL loss can be linearly interpolated together with traditional multitask (denoted as DMTL) loss, with a tunable interpolation factor. Experiments are run by augmenting different multitask learning methods with the new loss on different datasets, and result shows change in accuracy is positively correlated to the weight of interpolation factor on GMTL loss.


**Questions:**

(1) Under what conditions can we assume the existence of such a confounding factor, and why the experiments satisfy these conditions?
(2) How to extend this two-target system to multiple targets?

**Limitations:**

Limitations of this method are not addressed in the paper. One limitation can be it is hard to infer the linear interpolation factor in general tasks.

**Strengths And Weaknesses:**

Strengths:
(1) This seems to be the first paper addressing confounding factors that affect multitask targets.
(2) The GMTL objective is formulated with clear logic.
(3) Experiment results support the claims.
Weaknesses:
(1) It is unclear why the experiments satisfy the assumption, where some confounding factor u cause y and y’.
(2) The authors mention that they use two targets only “without loss of generality”, but it is unclear how to extend this method to multiple targets - do we assume a common confounding factor for all targets, or one confounding factor per pair of targets, or something else?

---

> ### Author Response · Authors · 2022-08-02
> **Response to Reviewer yfCS**
>
> We appreciate you finding our work to be novel, formulated with clear logic, and supported by experimental results. We address your questions below, where “R” is reviewer, and “A” is author.
>
> R: “It is unclear why the experiments satisfy this assumption, where some confounding factor $U$ cause $Y$ and $Y’$.”
>
> A: Our assumption that an unobserved $U$ causes $Y$ and $Y’$ implies $p(y, y’) = p(y, y’ \mid u) p(u)$, where both $p(y, y’ \mid u)$ and $p(u)$ are unknown. This means that $p(y, y’)$ can shift arbitrarily without restriction. This is precisely the setting of our experiments, since we evaluate across an unrestricted set of shifts in $p(y, y’)$.
>
> R: “Under what conditions can we assume the existence of such a confounding factor?”
>
> A: If there is no confounder, then $Y$ and $Y’$ are marginally independent. This can be tested with independence testing.
>
> R: “The authors mention that they use two targets only ‘without loss of generality,’ but it is unclear how to extend this method to multiple targets - do we assume a common confounding factor for all targets, or one confounding factor per pair of targets, or something else?”
>
> A: GMTL already works with more than two targets, and is agnostic to the number of confounders that cause them. All that matters is that the targets are conditionally dependent given the input. Regardless of how many targets or confounders there are, the procedure remains the same - estimate $p(y_1, \dotsc, y_n)$ jointly, and compute $\text{argmax}_{y_1, \dotsc, y_n} \log p(y_1 \mid x) + \dotsc + \log p(y_n \mid x) - \alpha \log p(y_1, \dotsc, y_n)$.
>
> As discussed in Section 3.3, the sparsity of the joint target space is a bottleneck of our method, since this complicates the accurate estimation of $p(y_1, \dotsc, y_n)$. Therefore, a large number of binary classification tasks can potentially present less difficulty than two tasks with a large joint target space, such as Taskonomy.

---

> > ### Author Response · Authors · 2022-08-09
> > **Response to Reviewer yfCS**
> >
> > Dear reviewer, could you confirm whether we addressed your concerns? If yes, please consider increasing our score.

---

### Official Review · Reviewer_q5bZ · 2022-07-11

**Rating:** 5
**Confidence:** 3
**Soundness:** 2 fair
**Presentation:** 3 good
**Contribution:** 3 good

**Summary:**

This paper proposes a generative multitask learning (GMTL) method to handle the existence of target-causing confounding from a generative model perspective. It tries to find the invariant information under different interventions (or in target-shift case) with the help of an auxiliary target. Experimental results show the robustness of the target shift.

**Questions:**

See my comments above.

**Limitations:**

Not available.

**Strengths And Weaknesses:**

Strengths:
1. This paper introduces the idea of anticausal learning for multitask learning. With the causal factorization of the joint distribution, the invariant information is used for tackling the target shift problem.
2. The accuracy of the experimental results demonstrates the effectiveness of the proposed methods.

Weaknesses:
1. Why p(y, y’|x) in Eq. (5) can be transferred into the product of p(y|x) and p(y’|x) (in Eq. (6)). According to Figure 1(c), in GMTL, y and y’ are not independent conditional on x, but are independent with each other conditional on latent variable U.
2. Is there any effective method to determine the $\alpha$ when there is no prior information?

---

> ### Author Response · Authors · 2022-08-02
> **Response to Reviewer q5bZ**
>
> Thank you for your positive feedback regarding the effectiveness of our method. We address your questions below, where “R” is reviewer, and “A” is author.
>
> R: “Why $p(y, y’ \mid x)$ in Eq. (5) can be transferred into the product of $p(y \mid x)$ and $p(y’ \mid x)$ (in Eq. (6)). According to Figure 1(c), in GMTL, $Y$ and $Y’$ are not independent conditional on $X$, but are independent with each other conditional on latent variable $U$.”
>
> A: See the first paragraph of Section 3.1, where we discuss that letting $p(y, y’ \mid x)$ factorize is an relaxation that we make purely out of convenience, and allows us to keep the training procedure the same as DMTL. The dependence between $Y$ and $Y’$ is captured solely by the $p(y, y’)$ term.
>
> We ran additional experiments on the Attributes of People dataset to show that our paper’s conclusions are not sensitive to this relaxation. We included these results in the revision and the supplementary material. On this dataset, we consider a pair of binary classification tasks, and model the joint target space as a four-way classification. We call the resulting architecture Full Parameter Sharing (FPS), since all weights are shared between the two tasks. Its results are generally comparable to the other MTL methods, which serves as evidence that the factorization in $p(y, y’ \mid x)$ is a sensible relaxation.
>
> R: “Is there any effective method to determine the $\alpha$ when there is no prior information?”
>
> A: See the third paragraph of Section 3.2, in which we describe a practical heuristic for selecting $\alpha$ in the absence of prior information. The parameter $\alpha$ represents a trade-off between in-distribution accuracy and out-of-distribution accuracy. Therefore, for maximum robustness, we specify a budget of accuracy that we can afford to lose on the in-distribution validation set, and choose the maximum $\alpha$ within that budget. Our results are in the supplementary materials Figures 1—4, and show that this heuristic can provide significant robustness improvements even for small budgets of in-distribution accuracy across multiple pairs of tasks, datasets, and MTL methods.

---

> > ### Comment · Reviewer_q5bZ · 2022-08-08
> > **Respond to authors' replies**
> >
> > Thank you for answering my comments.
> >
> > I still disagree that $p(y,y '|x)$ is rational to be factorized, because this paper mainly considers the MTL problem with target-causing confounder $U$. $U$ is an unobserved common cause of $y$ and $y '$. If we test the independence between $y$ and $y '$ only from observed data, $y$ and $y '$ are dependent, even when conditional on $X$. Is it possible to provide some theoretical analysis about the error with factorizing?
> >
> > Moreover, I am concerned that how to label the auxiliary targets $Y '$? Does it require additional manual labeling?

---

> > > ### Author Response · Authors · 2022-08-09
> > > **Response to Reviewer q5bZ**
> > >
> > > Thank you for your response. Under our causal graph, $Y$ and $Y’$ are indeed conditionally dependent given $X$. When viewed in isolation, us allowing $p(y, y’ | x)$ to factorize appears to violate this. It is important to understand our reason for letting $p(y, y’ | x)$ factorize – it isn’t because we think $Y$ and $Y$ are conditionally independent given $X$. We do it so that we can easily apply the GMTL inference objective using any existing MTL method, while skipping the difficult process of training MTL networks.
> > >
> > > GMTL’s inference objective is $\text{argmax}_{y, y’} \log p(x \mid y, y’)$. Using this inference objective breaks the spurious dependencies between the input and targets, and improves robustness to target shift. Since $p(x \mid y, y’)$ is difficult to estimate directly, we take several steps to make things easier. The first is to use Bayes rule to rewrite $\log p(x \mid y, y’)$ as $\log p(y, y’ \mid x) - \log p(y, y’)$. We then estimate $p(y, y')$, which is easy for categorical target since it just requires counting co-occurrences.
> > >
> > > This leaves us with estimating $p(y, y' \mid x)$. Instead of estimating this ourselves, we leverage existing MTL methods. Since training in MTL is notoriously difficult, it is very beneficial for us to be able to use existing methods. Since existing methods let $p(y, y' \mid x)$ factorize, we adopt this relaxation as well. To emphasize, we make this relaxation in order to leverage existing MTL methods, and not because we think $Y$ and $Y$ are conditionally independent given $X$.
> > >
> > > To give an example of what would be required if we didn't let $p(y, y' \mid x)$ factorize on Taskonomy, we would have to modify existing MTL methods to have a single output layer with dimension $100{\times}64$. We would then have to train this modified MTL network before we can apply GMTL.
> > >
> > > Even though we let the $p(y, y' \mid x)$ term factorize, if you consider our entire inference objective, the $p(y, y')$ term makes us consistent with the assumption that $Y$ and $Y$ are conditionally dependent given $X$. Our main contribution is to show that we can turn any existing MTL classifier into a robust classifier just by estimating $p(y, y')$ and performing inference jointly across all targets.
> > >
> > > As for labeling $Y’$, our setup is multitask learning, where datasets come labeled with multiple targets. How these datasets are labeled depends on the dataset, and this question is orthogonal to our method. For example, the object and scene labels in Taskonomy, which we used in our experiments, were automatically labeled. You can find details about the process here: https://github.com/StanfordVL/taskonomy/tree/master/data.

---

### Official Review · Reviewer_yUFE · 2022-07-20

**Rating:** 5
**Confidence:** 4
**Soundness:** 2 fair
**Presentation:** 3 good
**Contribution:** 2 fair

**Summary:**

In this paper, the authors propose a method for improving model robustness to target shift (i.e., shift in prior probability of labels) in multitask learning (MTL). Specifically, the authors consider a different hypothetical data generating process (DGP) than common MTL formulations: the authors assume that the observed data is generated causally from the targets, and that the targets are subject to unobserved confounding. This differs from the common assumption that tasks $Y$ and $Y'$ are conditionally independent given the data $X$. The authors use this DGP to show that modeling the the likelihood (i.e., $p(x|y,y')$) is invariant to shifts in the unobserved confounder, and thus is invariant to target shifts. Using Bayes rule and relaxing to factorize by task, the proposed method (GMTL) amounts to a modification of "standard" discriminative MTL to account for possible changes to the label distribution. Using synthetic and reweighted (to simulate target shifts) data, the authors show that with appropriate tuning the proposed method yields more robustness to target shifts that standard MTL which ignores the possibility of target shifts.

**Questions:**

Could the authors please address my questions about:
* What is the effect of allowing $p(y, y'|x)$ to factorize (when it does not under GMTL assumptions)? (See discussion above)
* Why is the $\alpha$ based interpolation the correct way to implement GMTL given that the uncertainty set is more general? What can be said about the model's performance across the uncertainty set? (See discussion above)

**Limitations:**

Yes, the authors adequately address the limitations and potential negative impact of their work.

**Strengths And Weaknesses:**

The idea in this paper is interesting: the authors propose to use knowledge about the data generating process (DGP) to improve the robustness of MTL under target shift. The premise of the approach (that a model of the data likelihood will be invariant to the shift) is sound and sensible, and the implementation is convenient (builds upon discriminate formulations to MTL). However, I think the framing of the target-causing confounding problem misses some context, and I have some questions/concerns about some choices/relaxations the authors make in implementing the proposed approach. In light of this, I think the paper is missing some more in-depth analysis of the performance expectations of the proposed approach.

**Re: Framing**

I have a few comments about the way the authors frame their method, as the framing seems to be presented as one of the primary contributions of the paper. In the introduction, the authors contrast the proposed approach (GMTL) with existing discriminative MTL (DMTL) approaches by saying that "DMTL is flawed under an alternative set of assumptions." I just want to point out that in Fig 1, from a statistical perspective, the graphs in Fig1a and Fig1b (DMTL under the different assumption sets) are incompatible. This is because the DMTL assumption that $Y \perp Y' | X$ holds in Fig1a but does not hold in Fig1b. Importantly, this is a *testable* assumption---because $X$, $Y$, and $Y'$ are observed in the training data, we can perform conditional independence tests to measure the compatibility of this assumption with the available data. Thus, in theory, it seems a user should be able to determine if the DMTL assumption is reasonable for their problem.

Second, while to my knowledge the GMTL formulation is novel for multitask learning, I would like to note that in supervised learning the DAG in Fig 1c has been considered in [1] (see Fig 1 in [1]). In one experiment they consider the target shift scenario (Section 5.1) and their approach ends up modeling (the equivalent of) $p(x|y,y')$. Given the similarities in the resulting objectives, I think some discussion of this work and its related threads (i.e., causal graphical approaches to stability/invariance to distribution shifts) is likely warranted.

I also think this connection to causal graphical approaches opens up additional methodological possibilities for the authors' line of work. For example, in Section 3, the authors note that the density estimation problem for $p(x | y, y')$ will, in general, be quite difficult and instead use equation (5) as their target of inference (and further relax it in eq (6)). However, by noting possible connections to the causal approaches, there may be alternative approaches to fitting the model such as inverse propensity weights (IPW). The authors would need to double check me, but I think reweighting the data by (something like) $\frac{1}{p(y, y')}$ during the training of a discriminative model would be an alternative approach for achieving the desired invariance (see, e.g., the weighting scheme in [2]).

[1] Subbaswamy, A., Schulam, P., & Saria, S. Preventing failures due to dataset shift: Learning predictive models that transport. AISTATS, 2019.

[2] Makar, M., Packer, B., Moldovan, D., Blalock, D., Halpern, Y., & D’Amour, A. Causally motivated shortcut removal using auxiliary labels. AISTATS, 2022.

**Re: relaxing the GMTL objective**

In Section 3.1, the authors "allow $p(y, y'|x)$ to factorize... purely out of convenience." While I understand why the authors want to do this (it reduces the multitask network part of GMTL to DMTL), it is clearly the case that $Y \not \perp Y' | X$ in Fig 1c (even if there were no unobserved confounder $U$ !). The authors assert that "we can allow this factorization as long as we capture the dependency between the targets when estimating $p(y, y')$", but this is not obvious to me. I would have expected to see either a theoretical justification or some experiments showing the effect of this relaxation by comparing the approach in eq (6) to one which does not assume that $p(y, y'|x)$ factorizes.

(As a side note, do cross-stitch networks (CSN) count as modeling $p(y, y'|x)$ jointly? If so, it might be possible to use CSN to perform the above comparison, but an approach that more explicitly jointly models $p(y, y'|x)$ might be better.)

**Re: interpretation of $\alpha$**

In equation (6), the authors also introduce a new hyperparameter $\alpha$ which interpolates between DMTL and GMTL. The motivation is to allow for solutions which generalize better under small or moderate shifts to the joint label distribution. However, it wasn't clear to me why this would be the correct or optimal approach for handling an uncertainty set of test distributions under target shift. In the simulated shifts considered in Section 5.4, it seems the authors allow for essentially arbitrary shifts to $p(y,y')$. This will include shifted label distributions that cannot be modeled in the form $p(y, y')^{1-\alpha}$, since the $\alpha$-based transformation will never change the mode (or relative order) of the probability mass function (except when $\alpha = 1$ in which case the label distribution becomes uniform).

The authors select an "optimal" $\alpha$ by finding the value that performs best on the shifted data, but why is the $\alpha$ based interpolation the correct way to implement GMTL given that the uncertainty set is more general? Can a given $\alpha$ value be framed as the solution to, e.g., a distributionally robust optimization problem for a certain uncertainty set? What can we say about the worst-case, average, or some other performance measure across the uncertainty set for GMTL? For $\alpha=1$, I see from eq (5) that predictions should be invariant to target shifts. What, though, can be said about model accuracy/performance? For $0 < \alpha < 1$ what can be said about performance?


**Minor notes**
* This is a very minor note, but isn't it the case that when interpreting $\alpha$ the test target distribution is *proportional* to $p(y,y')^{1-\alpha}$? If $p$ is a valid pmf, then $q=p^{1-\alpha}$ generally won't sum to 1 and needs to be renormalized.

---

> ### Author Response · Authors · 2022-08-02
> **Response to Reviewer yUFE**
>
> We are encouraged that you found the premise of our approach to be interesting, as well as sound and sensible. We address your comments and questions below, where "R" is reviewer, and "A" is author.
>
> R: “Given the similarities in the resulting objectives, I think some discussion of this work and its related threads (i.e., causal graphical approaches to stability/invariance to distribution shifts) is likely warranted.”
>
> A: We included Subbaswamy et al., 2019 in the related work of our revision. Their motivation is the same as us, which is to predict using intervention distributions to improve robustness to specific distribution shifts.
>
> R: “The authors would need to double check me, but I think reweighting the data by (something like) $\frac{1}{p(y, y’)}$ during the training of a discriminative model would be an alternative approach for achieving the desired invariance.”
>
> We included Makar et al., 2022 in the related work and in Section 3.2 of our revision. Indeed, our desired invariance can be achieved by weighting the training examples by $\frac{q(y, y’)}{p(y, y’)}$, where $q(y, y’)$ is the assumed distribution, and $p(y, y’)$ is the empirical distribution. Since we assume $q(y, y’) = p(y, y’)^{1 - \alpha}$, we would reweight the examples with $\frac{1}{p(y, y’)^\alpha}$.
>
> R: “I would have expected to see either a theoretical justification or some experiments showing the effect of this relaxation by comparing the approach in eq (6) to one which does not assume that $p(y, y’ \mid x)$ factorizes.”
>
> A: As you suggested, we ran additional experiments on the Attributes of People dataset to show that our paper’s conclusions are not sensitive to this relaxation. We added the results to the revision and supplementary material. On this dataset, we consider a pair of binary classification tasks, and model the joint target space as a four-way classification. We call the resulting architecture Full Parameter Sharing (FPS), since all weights are shared between the two tasks. Its results are generally comparable to the other MTL methods, which serves as evidence that the factorization in $p(y, y’ \mid x)$ is a sensible relaxation.
>
> R: “The motivation is to allow for solutions which generalize better under small or moderate shifts to the joint label distribution.”
>
> A: Our setup assumes that there can be arbitrary shifts to the target distribution, not just small or moderate ones. This is because $U$ determines $p(y, y’)$, and we have no knowledge about $p(u)$ or $p(y, y’ \mid u)$. This is reflected in our experimental setup, as we evaluate across arbitrary shifts to $p(y, y’)$.
>
> R: “This will included shifted label distributions that cannot be modeled in the form $p(y, y’)^{1-\alpha}$, since the $\alpha$-based transformation will never change the mode (or relative order) of the probability mass function.”
>
> A: In the first paragraph of Section 3.2, we said that “GMTL corresponds to assuming the test target distribution is $p(y, y’)^{1-\alpha}$.” This is easily misinterpreted, so we rewrote this paragraph in our revision. We are not assuming a specific form of the test target distribution. Instead, $\alpha$ represents our uncertainty about it. In the case of complete certainty, we set $\alpha = 0$ to use the empirical distribution. In the case of complete uncertainty, we set $\alpha = 1$ to completely remove the influence of the empirical distribution. Somewhere in the middle, we use $\propto p(y, y’)^{1-\alpha}$ where $\alpha \in (0, 1)$. In light of this, it is understood and acceptable that $p(y, y’)^{1-\alpha}$ cannot model arbitrary distributions.
>
> R: “The authors select an ‘optimal’ $\alpha$ by finding the value that performs best on the shifted data, but why is the $\alpha$ based interpolation the correct way to implement GMTL given that the uncertainty set is more general?”
>
> A: The $\alpha$ based interpolation is correct because it allows us to interpolate between the two extremes of complete certainty ($\alpha = 0$) and complete uncertainty $(\alpha = 1)$ regarding the test target distribution. The “optimal” $\alpha$ experiments show that our interpretation of $\alpha$ is correct.
>
> R: “Can a given $\alpha$ value be framed as the solution to, e.g., a distributionally robust optimization problem for a certain uncertainty set? What can we say about the worst-case, average, or some other performance measure across the uncertainty set for GMTL? For $\alpha = 1$, I see from eq(5) that predictions should be invariant to target shifts. What, though, can be said about model accuracy/performance? For $0 < \alpha < 1$ what can be said about performance?"
>
> A: Since we consider a completely unrestricted uncertainty set, it is not possible to make any performance guarantees. This is generally true, and is not a limitation of GMTL.
>
> R: Isn’t it the case that when interpreting $\alpha$ the test target distribution is proportional to $p(y, y’)^{1 - \alpha}$?
>
> A: Yes, we have corrected this in the revision.

---

> > ### Comment · Reviewer_yUFE · 2022-08-08
> > **Thanks for the responses, but perhaps some questions remain**
> >
> > Thanks to the authors for their responses to my comments/questions. I think the manuscript is clearly headed in the right direction with respect to improvements. For the reasons listed below I am inclined to maintain my score.
> >
> > Regarding factorizing $p(y, y')$, I think that from a correctness perspective it would be much easier for the authors if they did away with the relaxation. I think this is a sticking point for Reviewer q5bZ and me because the relaxation seems unnecessary (what is the tangible benefit from this relaxation?) and involves a conditional independence that simply does not hold given the author's assumed setup. If the empirical performance is comparable, why make the relaxation if you don't need it? I couldn't find the empirical results for the additional experiments, but if there is no real performance change, then from a theoretical perspective the method's soundness is improved by not making the relaxation which seems important! Why compromise on soundness if you don't need to?
> >
> > Regarding optimality from a distributionally robust perspective, I don't think you can wave this away by saying that it is an "unrestricted uncertainty set". I think it's outside the scope of my review to make some of the connections concrete for the authors, but it seems to me that decision theoretic results should allow one to establish that for the uncertainty set of all positive distributions $\mathcal{P}$ over $\mathcal{Y} \times  \mathcal{Y'}$, there exists a $Q \in \mathcal{P}$ that produces a predictor that is minimax optimal across $\mathcal{P}$. This would correspond to a loss function-specific maximum generalized entropy predictor. For details see Grunwald & David (2004).
> >
> > Grünwald, Peter D., and A. Philip Dawid. "Game theory, maximum entropy, minimum discrepancy and robust Bayesian decision theory." the Annals of Statistics 32.4 (2004): 1367-1433.
> >
> > And, as a possibly related example, in the supervised learning setting with general DAGs with latent variables, Subbaswamy et al. (2019) provide sufficient graphical conditions for when the $\alpha=1$ solution is minimax optimal for MSE in regression. I say all this because I genuinely believe the "optimality" component is an important part of the authors' story, and that there is much more that can be said here.

---

> > > ### Author Response · Authors · 2022-08-09
> > > **Response to Reviewer yUFE**
> > >
> > > Thank you for your response. There is a significant tangible benefit in letting $p(y, y’ | x)$ factorize, because it allows us to apply GMTL to MTL networks trained with existing methods. Without this relaxation, we would need to modify existing MTL methods to model the joint output space (instead of having task-specific heads), and train the modified networks. This is no easy task, since training in MTL is notoriously difficult. In the case of No Parameter Sharing (NPS), the relaxation allows us to combine two single-task networks into a robust MTL network.
> > >
> > > This is one of the key contributions of our paper – we show that it is sufficient to estimate $p(y, y’)$ and perform inference jointly over all targets in order to turn any existing MTL network (or multiple single-task networks) into a robust multitask classifier. To emphasize, with this relaxation GMTL becomes an inference-only procedure. Without the relaxation, GMTL additionally requires a training stage. The factorization also has an additional benefit in terms of evaluation in our paper, since it allows for a fair comparison between DMTL and GMTL.
> > >
> > > As for optimality, our motivation for using the uniform prior in the case of $\alpha = 1$ stems not from its optimality under certain assumptions, but because it corresponds to completely removing the influence of the empirical $p(y, y')$. We leave it to future work to investigate other alternatives to the uniform prior over under various assumptions and settings.
> > >
> > > You also mentioned not being able to find the empirical results for the additional experiments. Please see the red "FPS" curve in Fig.3 of the revised main text, and in most of the figures (all but the Taskonomy figures) in the revised supplementary material.

---

### Author Response · Authors · 2022-08-05
**Response to all reviewers**

We sincerely thank the reviewers for taking the time to read our paper, and for leaving thoughtful and constructive feedback. We are particularly encouraged by the following positive sentiment:

- The idea is interesting, the premise of the approach is sound and sensible, and the implementation is convenient. (Reviewer yUFE)
- The experimental results demonstrates the effectiveness of the proposed methods. (Reviewer q5bZ)
- The idea is novel, the method is formulated with clear logic, and the experimental results support the claims. (Reviewer yfCS)

Reviewers yUFE and q5bZ asked about our choice of letting $p(y, y’ \mid x)$ factorize. We performed a set of additional experiments to show that this relaxation does not change our conclusions, and included these results in the paper and supplementary material.

We provided detailed responses to the comments and questions of each reviewer. Please do not hesitate to ask for additional clarification or experimental results. Thank you again for your time.

---

### Meta-Review · Area_Chair_iPwp · 2022-08-28

**Recommendation:** Accept
**Confidence:** Certain

**Metareview:**

The decision is to accept the paper.

The paper presents an interesting perspective on confounding in the marginal distribution of task targets in multi-task learning settings, under an anti-causal prediction and no-unobserved-confounders assumption. The paper suggests an inference time strategy for eliminating some of the influence of the marginal correlation between task labels by subtracting off their joint log-probability from an inference-time objective. For convenience (and usability with current MTL training pipelines), the strategy employs an additional factorization in the posterior over tasks. The authors demonstrate effectiveness on several datasets.

The reviewers agreed that this was a fresh perspective on this problem, but also noted some limitations that could deserve more discussion in the paper. In particular, it would be useful for the authors to give an example (even a toy example) where the factorization really does hurt compared to an oracle that had access to the joint p(y, y' | x), which would motivate more fundamental research on the MTL side, and give end-users a sense of whether the method was appropriate for their problem.

Less pressingly, additional commentary on the particular modeling assumption about downstream domains that is being made with the alpha parameterization of the objective would also be useful; here, too, it would be useful to have a toy example where this simplification might also yield sub-optimal results, or at least a note that there may be other interpolation schemes between DMTL and GMTL that would work better. Basically, framing this as _a_ sensitive solution rather than _the_ solution would be helpful to point other researchers toward directions for future work.

Despite some of these reviewer concerns, I think there is enough here that the community would find this paper stimulating and useful.

**Award:**

No

---

### Decision · Program_Chairs · 2022-09-14

Accept